# Circular RNA encoded MET variant promotes glioblastoma tumorigenesis

Jian Zhong [1,2,12], Xujia Wu [1,2,12], Yixin Gao[1,2,12], Junju Chen[1,2,12], Maolei Zhang[1,2], Huangkai Zhou[1,2], Jia Yang [1,2], Feizhe Xiao[3], Xuesong Yang[1,2], Nunu Huang[1,2], Haoyue Qi[4,5], Xiuxing Wang [6,7,8,9] ✉, Fan Bai [10,11] ✉, Yu Shi [4,5] ✉ & Nu Zhang [1,2] ✉

Activated by its single ligand, hepatocyte growth factor (HGF), the receptor tyrosine kinase MET is pivotal in promoting glioblastoma (GBM) stem cell self-renewal, invasiveness and tumorigenicity. Nevertheless, HGF/MET-targeted therapy has shown limited clinical benefits in GBM patients, suggesting hidden mechanisms of MET signalling in GBM. Here, we show that circular MET RNA (circMET) encodes a 404-amino-acid MET variant (MET404) facilitated by the N[6]-methyladenosine (m[6]A) reader YTHDF2. Genetic ablation of circMET inhibits MET404 expression in mice and attenuates MET signalling. Conversely, MET404 knock-in (KI) plus *P53* knock-out (KO) in mouse astrocytes initiates GBM tumorigenesis and shortens the overall survival. MET404 directly interacts with the MET β subunit and forms a constitutively activated MET receptor whose activity does not require HGF stimulation. High MET404 expression predicts poor prognosis in GBM patients, indicating its clinical relevance. Targeting MET404 through a neutralizing antibody or genetic ablation reduces GBM tumorigenicity in vitro and in vivo, and combinatorial benefits are obtained with the addition of a traditional MET inhibitor. Overall, we identify a MET variant that promotes GBM tumorigenicity, offering a potential therapeutic strategy for GBM patients, especially those with MET hyperactivation.

Glioblastoma (GBM), the most common WHO grade IV primary brain cancer, is virtually incurable even with multimodal treatments, including surgery, radiotherapy and chemotherapy[1]. In the past decade, the median survival of GBM patients has remained approximately 12–15 months despite the utilization of latest approaches, including small molecule targeted therapy and immunotherapy[2–4]. GBM displays remarkable intratumoural heterogenicity, bearing multiple genetic or epigenetic alterations[5], such as loss of *PTEN* and *P53*; amplification of receptor tyrosine kinases (RTKs); inactivation of *CDKN2A* (*p16/INK4A*) and *CDKN2B*; and mutations in isocitrate dehydrogenase 1 and 2 (*IDH1/2*, a discriminant between primary and secondary GBM[6,7]). Of note, a high level of *MET* alteration is found in ~4% of GBM tumours[5,8]. A considerable proportion of GBM tumours with amplification of *EGFR* have aberrant *MET* expression[9], and activated

MET signalling is more commonly seen in secondary GBM[10]. Nevertheless, combined onartuzumab/bevacizumab targeted therapy displayed no benefit to GBM patients in phase II clinical trials[11], indicating the necessity of an in-depth study of MET signalling in GBM.

Recently, an increasing number of circular RNAs (circRNAs) have been shown to generate undiscovered critical molecular targets in human cancers, including GBM[12–15]. Driven by an internal ribosomal entry site (IRES) or N[6]-methyladenosine (m[6]A) modification, circRNAs can encode functional peptides or proteins to influence cancer stem cells (CSCs) in terms of self-renewal, invasion and therapeutic resistance[12,13,16,17]. Of note, the m[6]A reader YTHDF2 is reportedly highly expressed and maintains the oncogene characteristics of glioma stem cells (GSCs)[18], highlighting the potential of therapeutic targets generated from m[6]A-modified circRNAs.

In this work, we identify circular MET RNA (circMET) as a m⁶A-modified coding circRNA in GBM. Using genetic models, we demonstrate that circMET encodes a protein, MET404, that drives GBM tumorigenesis and reveal that the MET404 and MET β subunit form a chimeric MET receptor that constitutively activates downstream effectors independent of HGF stimulation. The combination of onartuzumab and MET404 antibody maximally inhibits mouse GBM xenograft progression and prolongs mouse overall survival, highlighting the future translational prospects of targeting MET404 in GBM.

## Results

### CircMET is a potential coding circRNA subjected to m⁶A modification

Many protein-coding circRNAs are driven by m⁶A modification[17,19]. To discover m⁶A-modified circRNAs, we performed RNA-seq and m⁶A-seq of 10 surgically resected GBM samples (Fig. 1a). The obtained reads were mapped to the ribosomal RNA database (Bowtie2[20]) and reference genome (HISAT2[21]). Twenty mers of the unmapped reads were collected and aligned to the reference genome to identify unique anchor positions within the splice site. Anchor reads that aligned in the

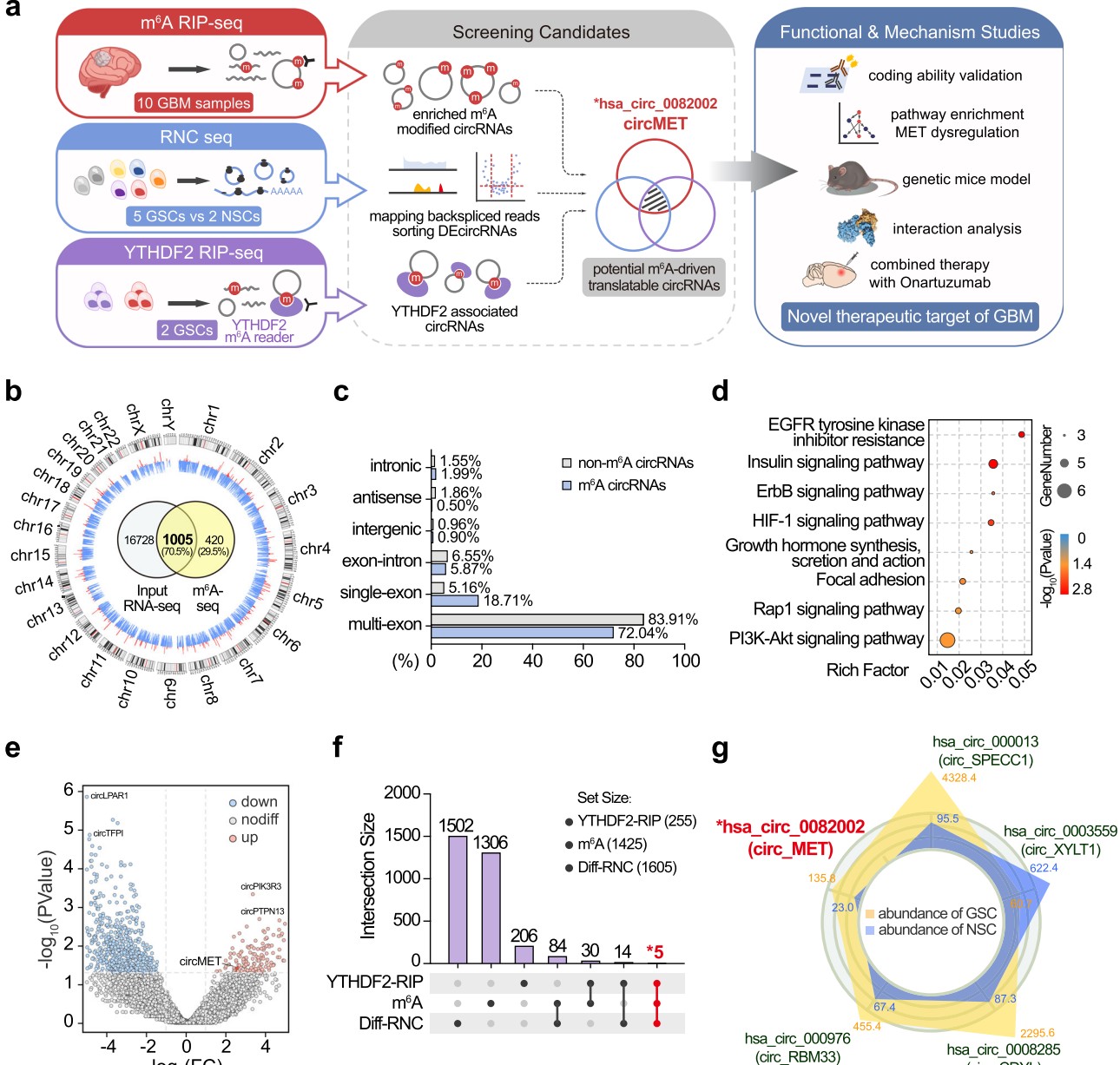

**Fig. 1 | CircMET is a potential coding circRNA subjected to m⁶A modification.** **a** Experimental strategy. GSC, glioma stem cell. NSC, neural stem cell. DEcircRNAs, differentially expressed circRNAs. **b** Circos plot shows the genome-wide m⁶A-modified circRNAs in GBM tissues. The outer circle represents the karyotype and centromere of chromosomes. The height of blue and red ticks represents the number of circRNAs detected (1-10) in input RNA-seq or m⁶A-seq, respectively. The Venn diagram in the centre illustrates the exact number of circRNAs detected in the m⁶A-seq and RNA-seq input data. **c** Proportion of the source regions of m⁶A circRNAs and non-m⁶A circRNAs. **d** KEGG enrichment analysis of the source genes of m⁶A circRNAs. Hypergeometric test. The exact P values are provided in Supplementary Data 2. **e** Volcano plot showing differentially expressed circRNAs between GSCs and NSCs identified from the RNC-seq data. **f** Upset plot illustrating the number of circRNAs detected by m⁶A-seq, RNC-seq and YTHDF2 RIP-seq. **g** Five candidate circRNAs were screened from the overlap of the m⁶A-seq, RNC-seq and YTHDF2 RIP-seq data. The yellow and blue polygons represent the RNC-seq expression levels of circRNAs in GSCs and NSCs, respectively. Source data are provided in the supplementary datasets.

reverse orientation (head-to-tail) indicated circRNA-characterized backsplicing and were subjected to find_circ[22] to identify circRNAs. A total of 1425 m[6]A-modified circRNAs were identified, and 1005 (70.5%) of these were found by total RNA-seq (17,733) (Fig. 1b and Supplementary Data 1). Compared with non-m[6]A-modified circRNAs, m[6]A-modified circRNAs were more likely generated from a single exon (18.71% vs. 5.16%) and were more highly expressed in cancerous tissues (Fig. 1c and Supplementary Fig. 1a). Notably, the source genes of m[6]A-modified circRNAs were enriched in multiple RTK pathways (Fig. 1d and Supplementary Data 2). Recent studies have revealed that m[6]A modifications can function as internal ribosome entry sites (IRESs), which facilitate binding between circRNAs and ribosomes, and thus promote their coding potential[17]. Next, we performed ribosome nascent chain sequencing (RNC-seq, capable of detecting circRNAs directly binding to ribosomes[23]) of five patient-derived GSCs and two control neural stem cells (NSCs) to reveal circRNAs with coding potential. Using find_circ, we identified a total of 1605 differentially expressed RNC-circRNAs ($\log_2$FC > 2, $P < 0.05$), with 349 upregulated and 1256 downregulated in GSCs (Fig. 1e and Supplementary Data 3). To verify the reliability of find_circ in identifying circRNAs, another algorithm (CIRIquant[24]) was also applied to annotate circRNAs in these sequencing data, and the results indicated that the output of both algorithms was highly consistent (Supplementary Fig. 1b). Among the 1425 m[6]A-modified circRNAs identified above, 89 were found to be differentially expressed between GSCs and NSCs by RNC-seq (Diff_RNC- m[6]A-circRNA). Binding to ribosomes does not ensure that an RNA can be translated[25]; however, YTHDF m[6]A readers reportedly facilitate the initiation of RNA translation[17,26]. Specifically, YTHDF2 is more strongly correlated with GBM prognosis and GSC maintenance compared with YTHDF1/3[18]. To further narrow down translatable circRNAs in GSCs, we next performed YTHDF2 RNA immunoprecipitation sequencing (RIP-seq) of GSC456 and GSC23 patient-derived GSCs and cross-referenced the data with the above m[6]A-seq and RNC-seq results. We detected a total of 255 YTHDF2-associated circRNAs (Fig. 1f and Supplementary Data 4), and five of these overlapped with the aforementioned 89 Diff_RNC-m[6]A-circRNAs. To here, we identified five candidate circRNAs, namely, circMET, circSPECC1, circXYLT1, circCDYL and circRBM33 (Fig. 1g). Given the critical role of MET in GBM tumorigenesis[9,10,27,28], we focused on circMET (hsa_circ_0082002) for further investigation.

## CircMET encodes the protein MET404

CircMET is formed from exon 2 of the *hMET* gene (Fig. 2a). Its counterpart in mice is formed from exon 3 of *mMET* (Supplementary Fig. 2a). The sequence of circMET is highly conserved among humans, Macaca, mice and rats (Supplementary Fig. 2b). We validated circMET (1214 nt) by using junction-specific primers (Fig. 2a) and tested its circRNA characteristics, including resistance to RNase R digestion (Supplementary Fig. 2c), inability to be amplified by oligo dT primer due to the absence of a poly-A tail (Supplementary Fig. 2d), a longer half-life (Supplementary Fig. 2e), cellular localization (Supplementary Fig. 2f) and relative expression in several GBM cells and GSCs (Supplementary Fig. 2g). CircMET is a cytoplasm-localized circRNA (Supplementary Fig. 2f, i) and is more highly expressed in GSCs compared with NSCs and normal human astrocytes (NHAs) (Supplementary Fig. 2g). The expression of circMET was also validated by Northern blotting using junction- and exon-specific probes (Supplementary Fig. 2j). By m[6]A-IP, we further validated the multiple m[6]A modifications of circMET (Fig. 2b), in accordance with the TransCirc database[29] (Supplementary Fig. 3a). Ribosome fragment analysis showed that circMET was detected in the heavy polysome fraction, which also supported the translational potential of circMET (Supplementary Fig. 3b). Furthermore, circMET contains a cross-species conserved ORF, which encodes a 404-amino-acid (a.a.) protein in both humans and mice (Fig. 2b, left and Supplementary Fig. 3c, d). This ORF uses the

same start codon as linear MET and spans the backsplicing junction site before meeting the stop codon, which generates a unique four-amino-acid-long C-terminus (INLS). We termed this undefined protein as MET404, and a MET404 antibody targeting the C-terminus was also synthesized and tested (Supplementary Fig. 3c, e). MET404 is a membrane- and cytoplasm-localized protein, as determined by immunofluorescence (IF) and cell fractionation immunoblotting (IB) (Supplementary Fig. 3f, g). Using whole-cell lysates and membrane proteins of GSC23 cells, we validated 100% of the amino acids in MET404, including all four specific C-terminal a.a., by liquid chromatography/mass spectrometry (LC/MS) (Supplementary Fig. 3h). To further confirm that MET404 was generated from circMET instead of *MET* linear splicing variants, we applied the CRISPR/Cas9 system to generate circMET knockout (KO) mice using a previously reported circRNA KO strategy[30] (Fig. 2c and Supplementary Fig. 4a). Deletion of the downstream intron sequence of *mMET* exon 3 did not alter *mMET* mRNA expression (Supplementary Fig. 4b). However, CRISPR/Cas9-induced downstream intron element deletion repressed the circularization efficiency and markedly decreased the circMET levels (Fig. 2d). Although circMET could not be completely deleted by this strategy, the markedly diminished MET404 expression and the unaffected MET expression in multiple organs of circMET KO mice further indicated that circMET encoded this protein (Fig. 2e).

## M[6]A modification and the YTHFD2 m[6]A reader are essential for circMET translation

CircMET is a m[6]A-modified, YTHDF2-associated circRNA and YTHDF2 is a validated glioma prognostic marker highly expressed in GBM[18]. To investigate whether YTHDF2 is essential for circMET translation, we knocked down (KD) YTHDF2 in GSC456 and GSC23 cells. YTHDF2 KD did not affect circMET expression but reduced the MET404 protein levels (Fig. 2f and Supplementary Fig. 4c). Moreover, the YTHDF2/circMET interaction was further investigated by RIP (Fig. 2g). Interestingly, knockdown of YTHDF1/3 did not affect MET404 expression (Supplementary Fig. 4d, e). Knockdown of the m[6]A writer METTL3 and overexpression (OE) of the m[6]A eraser ALKBH5 also reduced the MET404 protein levels, whereas the same results were not obtained with the ALKBH5 H204A mutant[31] (Supplementary Fig. 4f, g). The YTHDF2 mutants W432A and W486A[32] could not interact with circMET as YTHDF2 WT did and thus did not alter MET404 protein expression (Fig. 2h and Supplementary Fig. 4h). Inspection of several clinical GBM samples and paired normal brain (NB) samples revealed that MET404 tended to be overexpressed in cancerous tissues, despite the absence of MET upregulation in some of the cases (Fig. 2i and Supplementary Fig. 4i). Compared with MET, MET404 was expressed at substantially comparable or even higher levels in some GSCs and GBM cancerous tissues, as detected with an MET N-terminal antibody that recognizes both MET and MET404 protein (Fig. 2j and Supplementary Fig. 4j, k). The high expression of MET404 may be attributed to the longer half-life of circMET (Supplementary Fig. 2e) and the rich m[6]A modification that can be recognized by YTHDF2, which facilitates highly efficient translation. Immunohistochemistry (IHC) of GBM samples revealed that MET404 expression was spatially correlated with YTHDF2 expression (Fig. 2k), and high MET404 expression predicted worse overall survival of GBM patients (Fig. 2l). Notably, the circMET and MET404 expression levels were markedly higher in GSCs than in NSCs or established GBM cell lines (Supplementary Figs. 2g, and 4l). Accordingly, circMET and MET404 were also upregulated in CD133-positive cells compared with CD133-negative cells derived from clinical surgical tumour samples (Supplementary Fig. 4m–p).

## MET404 promotes GBM tumorigenesis by activating MET signalling

MET404 localized mainly on the cell membrane (Supplementary Fig. 3g). To explore the unknown function of MET404, we performed

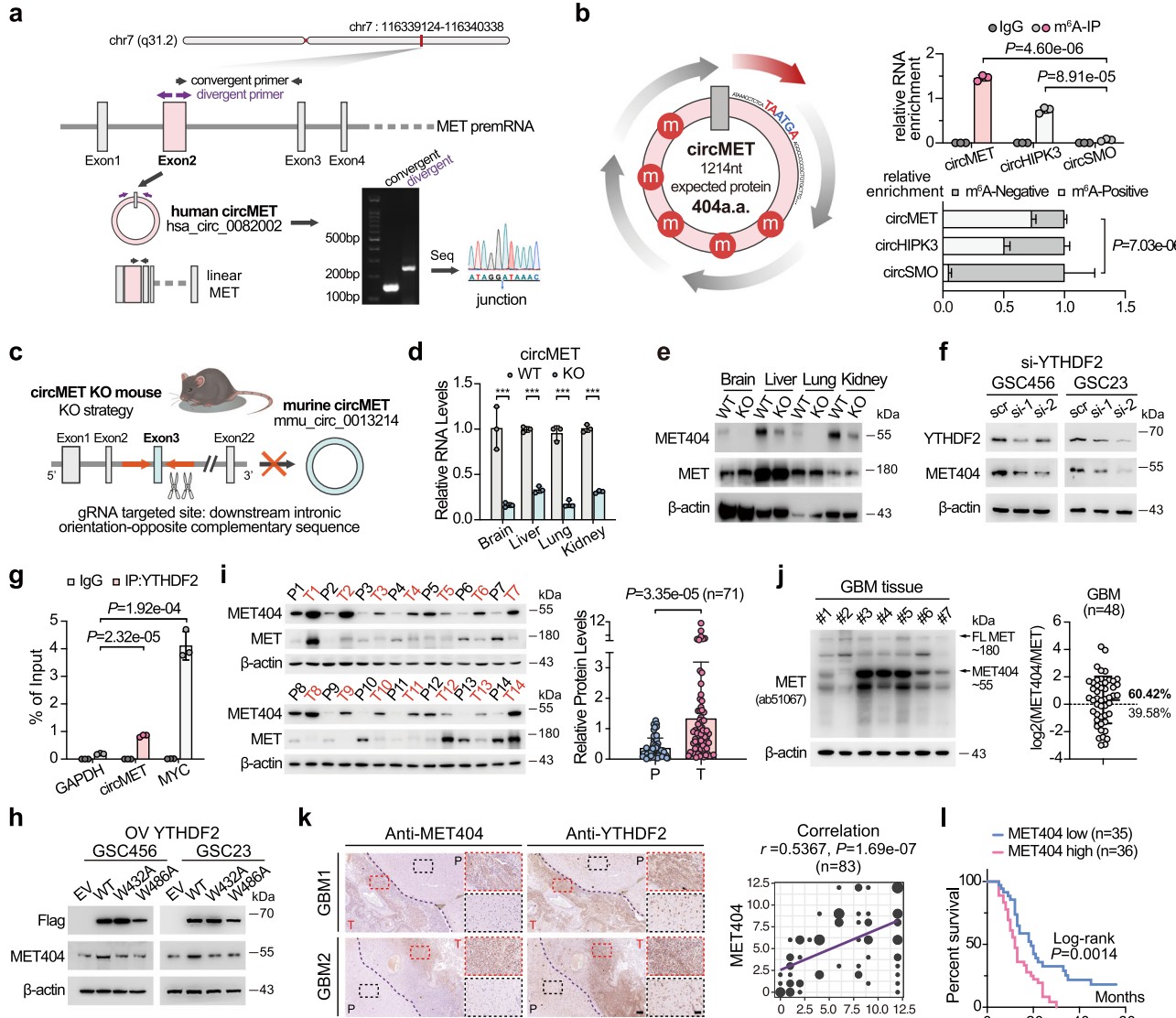

**Fig. 2 | CircMET encodes the protein MET404 driven by the m⁶A reader YTHDF2. a** Illustration of the annotated genomic region of human circMET, different splicing forms of MET premRNA, and the validation strategy for the existence of circular MET exon 2. Convergent (black) and divergent (purple) primers were designed to amplify the linear or back-spliced products. PCR analysis followed by Sanger sequencing using the indicated divergent primers revealed the "head-to-tail" splicing of human MET exon 2. **b** Left, illustration of the ORF and m⁶A modifications of circMET. Upper right, GSC23 cells were subjected to N⁶-methyladenosine immunoprecipitation (m⁶A IP) using an anti-m⁶A antibody or IgG control. The immunoprecipitates were analysed by RT-qPCR with specific primers for circMET, circHIPK3 (positive control) and circSMO (negative control). *n* = 3 independent experiments. M⁶A-IP, circMET vs circSMO, *P* = 4.60e−06; circHIPK3 vs circSMO, *P* = 8.91e−05. Lower right, RNAs from the eluate (m⁶A positive) and supernatant (m⁶A negative) after m⁶A immunoprecipitation were purified and subjected to subsequent RT-qPCR analysis using the indicated primers. *n* = 3 independent experiments. circMET vs circSMO, *P* = 7.03e−06. **c** Schematic strategy of the circMET knock-out (KO) mouse model using CRISPR/Cas9 technology. Guide RNAs (gRNAs) were designed to target downstream introns containing orientation-opposite complementary sequences that facilitate the circulation of exon 3 of mouse MET premRNA. **d** CircMET RNA levels in the indicated organs of the circMET KO mouse model. *n* = 3 independent experiments. Brain, *P* = 3.46e−03; Liver, *P* = 1.90e−05; Lung, *P* = 1.24e−04; Kidney, *P* = 1.08e−05. **e** Protein levels of MET404 and MET in the indicated organs of the wild-type (WT) and circMET KO mouse models. Representative of three independent experiments. **f** Protein levels of MET404 in YTHDF2-siRNA-transfected GSC456 and GSC23 cells. Representative of three independent experiments. **g** GSC23 cells were subjected to RNA immunoprecipitation using an anti-YTHDF2 antibody or IgG control and subsequent RT−qPCR analysis with specific primers for circMET, MYC (positive control) and GAPDH (negative control). *n* = 3 independent experiments. IP YTHDF2, circMET vs GAPDH, *P* = 2.32e−05; MYC vs GAPDH, *P* = 1.92e−04. **h** Protein levels of MET404 in GSC456 and GSC23 cells transfected with empty vector (EV), wild-type (WT) YTHDF2, or mutated YTHDF2 (W432A and W486A). Representative of three independent experiments. **i** Left, representative expression levels of MET404 and MET in 14 randomly selected independent paired GBM samples. P, peritumour tissue. T, tumour. Right, semiquantitative analysis of MET404 expression levels based on immunoblot greyscale analysis in a cohort of 71 independent GBM samples. Two-tailed paired *t* test, *P* = 3.35e−05. **j** Left, representative expression levels of MET and MET404 in 7 randomly selected independent GBM cancerous tissues detected using an antibody that simultaneously recognizes MET and MET404. Right, semiquantitative analysis of the ratio of MET404 to MET expression levels based on immunoblot greyscale analysis (*n* = 48). **k** Left, representative immunohistochemical (IHC) images showing the spatial correlation of MET404 and YTHDF2 from two GBM samples. The dark purple dashed lines indicate the border of the tumour. P, peritumour tissue. T, tumour. Scale bar, 250 μm. The magnified view is shown inside the dashed box. Scale bar, 100 μm. Right, correlation analysis of MET404 and YTHDF2 expression levels based on IHC scores obtained from a cohort of 83 independent GBM samples. *r* = 0.5367, *P* = 1.69e−07. **l** Survival analysis of the aforementioned GBM cohort of 71 patients stratified by MET404 expression (with median expression level as the cut-off value). Log-rank test, *P* = 0.0014. The data are presented as the mean ± SD. Unpaired two-tailed Student's *t* test was used to determine the significance of the differences between the indicated groups where applicable. *P < 0.05; **P < 0.01; ***P < 0.001. Source data are provided as a Source data file.

RNA-seq of GSC23 cells expressing circMET shRNA or scramble shRNA and of GSC28 cells overexpressing circMET RNA or empty vector. KEGG enrichment analysis was performed, and 31 pathways, including several receptor tyrosine kinase (RTK)-related pathways and tumour-related pathways (Fig. 3a and Supplementary Data 5), were significantly enriched in both comparison groups, implying that MET404 is involved in multiple cancerous signalling pathways. In stable circMET KD GSC456 and GSC23 cells, we examined several RTKs, including EGFR, PDGFRα, MET, FGFR and TrkB, and found that only p-MET was markedly inhibited (Fig. 3b and Supplementary Fig. 5a, b). In addition, p-MET downstream activation (p-AKT, p-ERK) was restrained. Cellular proliferation, limited dilution assays and neurosphere formation ability assays revealed inhibition in both types of circMET KD GSCs (Fig. 3c–f and Supplementary Fig. 5c–e). OE of circMET RNA or the MET404 ORF in GSC28 cells increased the p-MET and downstream signal levels and the abovementioned biological functions (Supplementary Fig. 5f–m). OE of a mutant circMET RNA (inserted with several adenine bases to disrupt the MET404 ORF) did not achieve the same effects in GSC28 cells, suggesting that these effects were dependent on MET404 rather than circMET RNA (Supplementary Fig. 5i–m). In a mouse in situ tumorigenesis assay, KD of MET404 inhibited mouse brain tumour xenograft growth and prolonged overall survival (Fig. 3g, h and Supplementary Fig. 5n), whereas OE of circMET RNA or MET404 ORF (but not circMET mut) further enhanced GBM progression (Supplementary Fig. 5o, p). Although homozygous MET KO mice could experience embryonic lethality[33], we did not observe abnormal phenotypes in circMET KO mice. This finding was probably observed because the intron KO reduced the circularization efficiency but did not completely abolish MET404 expression and had no effect on MET expression. However, the challenging of circMET KO mice with CCl$_4$[34] inhibited the liver repair process compared with that in WT mice (Supplementary Fig. 5q, r). The MET signalling intensity, which is critical for liver regeneration[35,36], was markedly lower in circMET KO mice than in WT mice (Supplementary Fig. 5s). Notably, CCl$_4$ treatment induced MET404 expression within 6 h, which suggested that MET404 responds rapidly and is critical for MET signalling (Supplementary Fig. 5s).

A previous report showed that P53$^{f/f}$; PTEN$^{f/-}$ mice developed spontaneous GBM[37]. Given that MET404-activated p-AKT could mimic PTEN$^{f/-}$, we generated P53$^{f/f}$; R26-MET404-flag$^{LSL/+}$; GFAP-Cre transgenic mice (MET404 KI mice, Fig. 4a–c). Most MET404 KI mice had developed brain tumours at 20 weeks (displayed central nervous system symptoms including paralysis, seizure and/or ataxia), and the tumour phenotypes were very similar to those of GBM, with highly activated MET signalling (Fig. 4d, e). Compared with that in control mice (P53$^{f/f}$; R26-MET404-flag$^{+/+}$; GFAP-Cre, MST 48.35 weeks, often died due to mouse or subcutaneous tumours), the overall survival was remarkably shorter in MET404 KI mice (MST 29.25 weeks, all died of CNS tumours) (Fig. 4f). The above results suggested that MET404 is critical for GBM tumorigenesis by activating MET signalling.

## MET404 activates the MET receptor by interacting with the MET β subunit

We next explored the mechanism by which MET404 activates MET signalling. Analysis of amino acid sequences strongly suggested that MET404 has a secretion signal peptide (Supplementary Fig. 6a, b). Indeed, IB of the supernatant from GSCs indicated that MET404 was secreted into the GSC culture medium (Fig. 5a). In the cerebrospinal fluid of GBM patients, the MET404 concentration fluctuated at 711.2 ± 217.5 pg/mL, as detected by sandwich enzyme-linked immunosorbent assay (ELISA) (Supplementary Fig. 6c). Because the exact effective in vivo concentration of MET404 at the tumour site is unmeasurable based on current methods, we applied gradient stimulation using the self-purified MET404 and defined the optimal in vitro stimulating concentration as 1 μg/mL (Supplementary Fig. 6d). Purified

MET404-activated MET and downstream signalling, and these effects could be antagonized by the addition of a MET404 neutralization antibody (Fig. 5b, c). The stably KD of the MET receptor in GSC28 and GSC387 cells inhibited the activation of MET and downstream signalling induced by purified MET404 (Fig. 5d). The abovementioned results suggested that MET404 exerted its effects through the MET receptor.

We then performed MET404 immunoprecipitation-mass spectrometry (IP-MS) and identified 196 candidate proteins as potential binding partners of MET404 (Supplementary Data 6). Enrichment analysis revealed several Gene Ontology (GO) terms related to extracellular localization and molecule binding (Fig. 5e), consistent with the role of MET404 as a secreted protein. Notably, MET was listed in the identified candidates (Fig. 5f and Supplementary Fig. 6e). Moreover, Reactome pathway analysis revealed that MET404-interacting proteins were highly associated with the RTK pathway, especially with MET signalling, including the regulation of MET activity, RAF/MAP kinase cascade, and MAPK family signalling cascades (Fig. 5g and Supplementary Data 7). By utilizing the STRING database, a protein interaction network with MET as the hub was established based on the identified MET404-binding candidates (Fig. 5h), and this network included 3 direct binding partners of MET (RPS27A, HSP90AA1 and JAK1) and another 42 indirect interacting proteins. Next, we validated the MET404/MET interaction in GSC23 cells by mutual IP (Fig. 5i). Immunofluorescence (IF) also supported the conclusion that MET404 and MET colocalized in GSC23 and GSC456 cells, particularly on the cell membrane (Fig. 5j).

To elucidate MET404/MET receptor binding sites, we employed the ClusPro server[38] to simulate a complex formed from MET404 and the MET ectodomain. The contact list of the final stable complex revealed that residues 542–650 (spanning the PSI and IPT1 domains of MET[39]) on MET were key to mediating its interaction with MET404 (Fig. 6a and Supplementary Table 1). Inspired by the analysis, we synthesized constructs of the full-length MET receptor and several truncated MET receptor domains to identify the exact domain on MET that interacts with MET404. Using eukaryotic purified MET404 and different MET truncations, we showed that MET404 interacted with the PSI + IPT1 domain of the MET receptor (Fig. 6b). In contrast, HGF interacts with the SEMA domain of the MET receptor[39]. Furthermore, we mutated several residues on MET404 that were predicted to have a strong interaction energy and found that both single mutations of Q328 and D358 sufficiently compromised the interaction (Supplementary Fig. 7a). Moreover, treatment with purified MET404 (1 μg/mL) activated p-MET and downstream effectors earlier than treatment with HGF (100 ng/mL) (Supplementary Fig. 7b), and treatment with the combination of MET404 and HGF exerted a synergistic effect on p-MET activation (Fig. 6c).

Due to the different interaction region to MET receptor, we hypothesized that MET404 could independently activate p-MET without HGF. We generated HGF KO GSC456 and GSC23 cells (Supplementary Fig. 7c). In both HGF KO cell lines, slight p-MET downregulation was observed (Supplementary Fig. 7d), suggesting that HGF is responsible for part of the p-MET activity. The addition of purified MET404 boosted p-MET and downstream signalling, indicating that MET404 alone could activate the MET receptor (Fig. 6d). The activation of RTKs requires dimerization upon ligand binding[40]. We next asked whether MET404 alone can induce dimerization of the MET receptor. We cotransfected constructs into 293 T HGF KO cells (Supplementary Fig. 7e) to express HA or flag-tagged MET monomer and performed coIP assays using the corresponding tag antibody. Interaction between differently tagged MET was observed in the control cells (indicating dimerization), and was strengthened by the addition of purified MET404 (Fig. 6e). To further prove the dimerization induced by MET404, we applied the NanoBit System[41]. Large Bit (LgBit) and Small Bit (SmBit) residues were fused to the MET monomer.

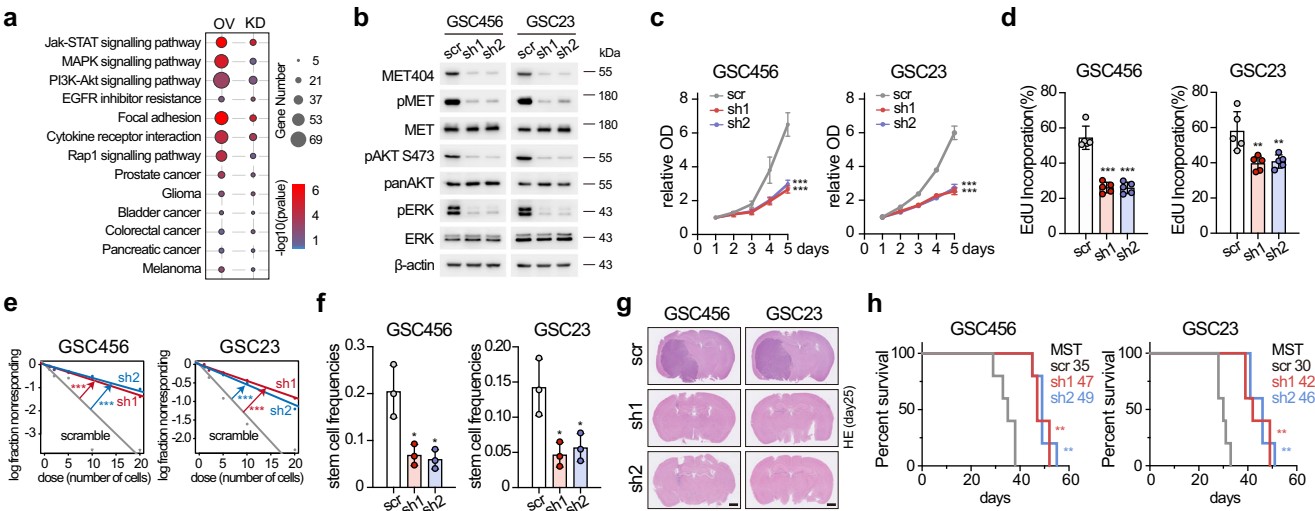

**Fig. 3 | MET404 promotes GBM tumorigenesis by activating MET signalling.**
**a** Bubble plot of the KEGG enrichment analysis results based on bulk RNA-seq data of GSCs with stable knockdown (KD) or overexpression (OE). *n* = 3 biologically independent samples for each group. **b** Protein levels of phospho-MET and downstream phospho-AKT (S473) and phospho-ERK signals in stable circMET KD cells. Representative of three independent experiments. **c** Proliferation of control and circMET stable KD GSC456 and GSC23 cells. *n* = 5 independent experiments. Two-way ANOVA test. GSC456, scr vs sh1, *P* = 3.0e−15; scr vs sh2, *P* = 1.8e−14. GSC23, scr vs sh1, *P* < 1.0e−15; scr vs sh2, *P* < 1.0e−15. **d** Quantification of the EdU incorporation assay of the control and stable circMET KD GSC456 and GSC23 cells. *n* = 5 independent experiments. GSC456, scr vs sh1, *P* = 2.3e−05; scr vs sh2, *P* = 2.4e−05. GSC23, scr vs sh1, *P* = 0.0081; scr vs sh2, *P* = 0.011. **e** Limited dilution assay (LDA) analysis of the control and stable circMET KD GSC456 and GSC23 cells. *n* = 3 independent experiments. GSC456, scr vs sh1, *P* = 4.78e−07; scr vs sh2, *P* = 1.88e

−08. GSC23, scr vs sh1, *P* = 1.05e−06; scr vs sh2, *P* = 3.64e−05. **f** Stem cell frequencies of LDA analysis of the control and stable circMET KD GSC456 and GSC23 cells. *n* = 3 independent experiments. GSC456, scr vs sh1, *P* = 0.0176; scr vs sh2, *P* = 0.0136. GSC23, scr vs sh1, *P* = 0.0194; scr vs sh2, *P* = 0.0298. **g** Representative haematoxylin and eosin (H&E)-stained brain slices from mice intracranially injected with the control and stable circMET KD GSC456 and GSC23 cells (*n* = 5 per group). Scale bar, 1 mm. **h** Survival analysis of mice intracranially injected with the control and stable circMET KD GSC456 and GSC23 cells (*n* = 5 per group). Log-rank test. GSC456, scr vs sh1, *P* = 0.0043; scr vs sh2, *P* = 0.0043. GSC23, scr vs sh1, *P* = 0.0017; scr vs sh2, *P* = 0.0017. The data are presented as the mean ± SD. The statistical tests are two-sided unless otherwise specified. Unpaired Student's *t* test was used to determine the significance of the differences between the indicated groups where applicable. *P* < 0.05; **P* < 0.01; ***P* < 0.001. Source data are provided as a Source data file.

Dimerization of MET results in approximation of these residues, and thus generates a luminescent signal. A marked signal was observed soon after the addition of MET404 in the HGF KO cells (Fig. 6f), demonstrating the ability of MET404 alone to induce MET dimerization and activation. In addition, we also generated MET404 KO GSC23 and GSC456 cells by abolishing circMET expression (Supplementary Fig. 7f). MET404 KO dramatically reduced the p-MET levels in GSCs, indicating that MET404 is a major MET activator in GSCs (Supplementary Fig. 7d). The addition of HGF to MET404 KO cells also increased the p-MET and p-AKT levels (Supplementary Fig. 7g). Although MET404 shares some sequences with linear MET at the N-terminus where the HGF-interacting SEMA domain is situated, interaction between HGF and MET404 was not observed in 293 *MET*-KO cells transfected with HGF-HA and MET404-His (Supplementary Fig. 7h, i). This finding was likely obtained due to the incomplete composition of the SEMA domain on MET404, which lacks key interfaces mediating the binding with HGF[39,42]. Taken together, these data corroborated that HGF and MET404 independently activate the MET receptor and synergize to maximize MET signalling.

Finally, we generated MET exon2 KO GSCs (Supplementary Fig. 7j). Given that the MET receptor is formed by α and β subunits, both of which are generated by cleavage of full-length MET[43], MET exon2 KO abolished both MET receptor and MET404 expression (Supplementary Fig. 7j). Treatment with HGF or MET404 in MET exon2 KO GSCs did not activate downstream p-AKT and p-ERK (Fig. 6g). We then re-expressed the β subunits in these cells and stimulated them with HGF or MET404. HGF stimulation could not activate p-MET and downstream signalling in these modified cells, supporting the conclusion that the α subunit is needed for the classic MET working model. In contrast, MET404 alone stimulated p-MET and p-AKT in these

modified cells, indicating that MET404 could form a complex with the β subunit and activate the MET receptor without HGF stimulation (Fig. 6g). Given the aberrant expression of MET404, our data suggested that MET404 is the key activator of hyperactivated MET signalling in GBM.

## Synergistic effects of targeting MET404 and onartuzumab in restraining GBM progression

In GBM clinical samples, we observed marked MET404 and p-MET colocalization and found that the MET404 intensity correlated with the p-MET levels (Fig. 7a). Furthermore, the p-MET levels were better correlated with MET404 expression in GBM than with MET receptor expression (Fig. 7b), supporting our previous conclusion that MET404 is the major activator of p-MET in GBM. The dose-dependent addition of MET-404 antibody gradually decreased the p-MET levels in GSC23 and GSC456 cells (Fig. 7c), implying the clinical potential of targeting MET404 in GBM. Notably, the combination of MET404 antibody and onartuzumab, an FDA-approved MET signalling inhibitor that showed a negative effect in GBM clinical trials[11], maximally inhibited MET signalling and downstream events including AKT/ERK activation and cellular viability (Fig. 7d). In mouse brain tumour xenografts, onartuzumab and MET404 antibody treatment achieved partial inhibition of MET phosphorylation and tumour progression; the combination of these two approaches, however, maximally inhibited GBM progression as well as in vivo MET signalling and prolonged the overall survival of the experimental mice to more than 60 days (MST 56−59 days, Fig. 7e−g). Compared with the small molecule RTK inhibitor crizotinib, the antibody combination also exerted better effects in arresting the growth of xenografts and prolonging mouse survival (Supplementary Fig. 8).

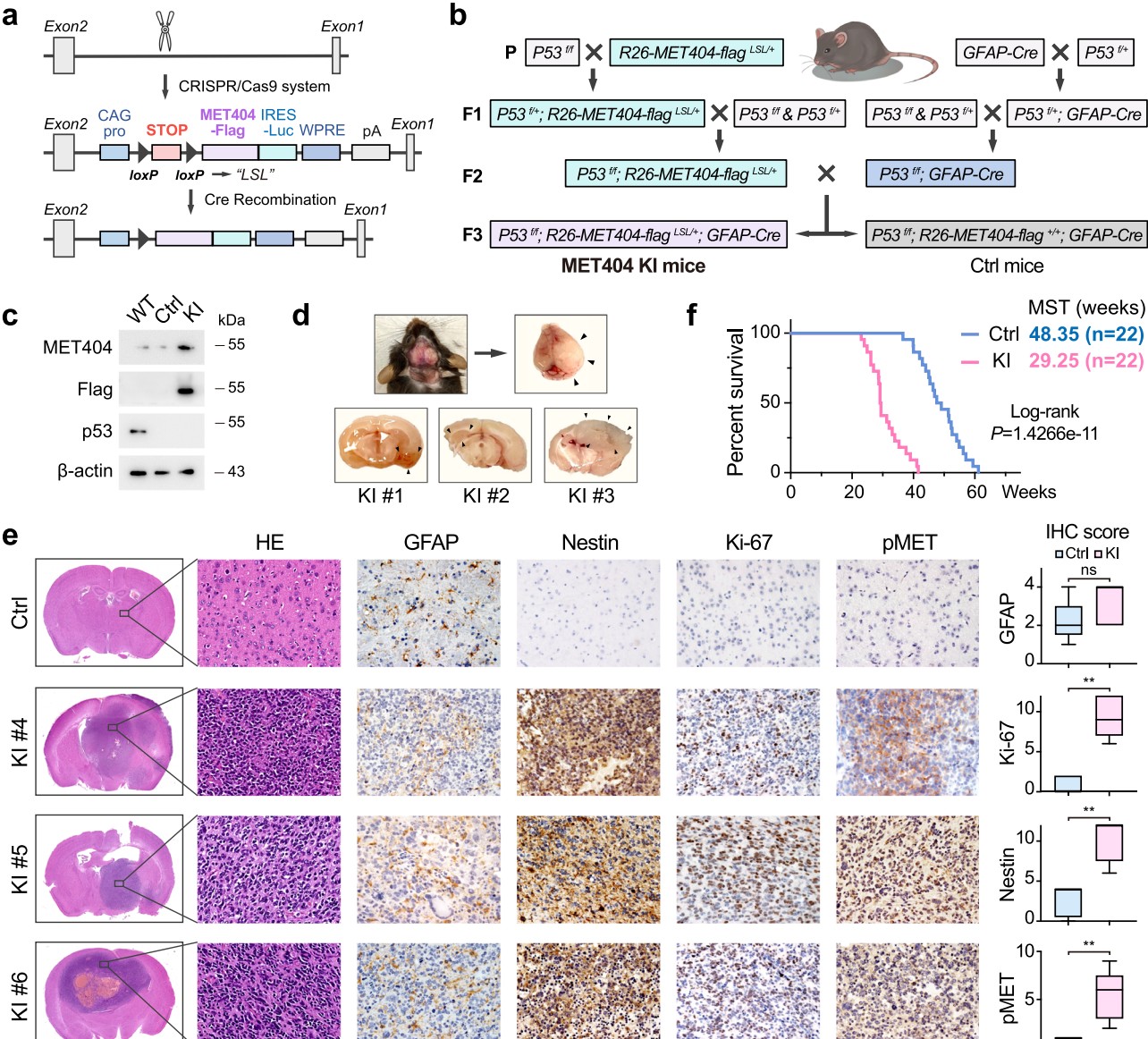

**Fig. 4 | MET404 conditional knock-in initiates GBM in a genetic mouse model.**
**a** Schematic illustration of CRISPR/Cas9 system-mediated knock-in (KI) of MET404 at the mouse Rosa26 locus. **b** Illustration of the crossing strategy of the GBM spontaneous mouse model with conditional MET404 KI (MET404 KI mouse, *P53^f/f^; R26-MET404-flag^LSL/+^; GFAP-Cre*) and control mice (*P53^f/f^; R26-MET404-flag^+/+^; GFAP-Cre*). "+" indicates the wild-type allele, and "*f*" denotes the conditional allele. **c** Immunoblotting of MET404-Flag and p53 in brain tissues from the indicated mice. Representative of three independent experiments. **d** Upper left, representative photograph of a MET404 KI mouse with a middle incision in its skull. Upper right, the swollen brain removed from the skull of the mouse on the left. Bottom, representative brain slice photograph from three MET404 KI mice. The black

arrows denote the edge of the tumour mass. **e** Left, representative H&E-stained and IHC images of MET404 KI and control mice using anti-GFAP, anti-Nestin, anti-Ki-67 and anti-phospho-MET antibodies. Scale bar, 1 mm (for H&E-stained whole brain slices); 50 μm (for enlarged H&E-stained brain slices and IHC images). Right, IHC scoring of the indicated proteins in randomly selected microscopy fields of each IHC image (*n* = 5 independent mouse samples). The data are presented as box plots containing the median (centre line), the first and third quartiles (box limits). The whiskers indicate the maxima and minima. Two-sided Mann–Whitney test. GFAP, *P* = 0.3651; Ki-67, *P* = 0.0079; Nestin, *P* = 0.0079; pMET, *P* = 0.0079. **f** Survival analysis of conditional MET404 KI and control mice (*n* = 22 per group). Log-rank test, *P* = 1.4266e−11. Source data are provided as a Source data file.

## Discussion

Recent high-throughput m[6]A-seq data indicated that at least 13% of circRNAs carry the m[6]A modification[17]. M[6]A modification is sufficient to initiate circRNA translation with the initiation factor eIF4G2 and m[6]A reader proteins such as YTHDF3[17]. Given that YTHDF2 is a more highly expressed m[6]A reader than YTHDF3 in GBM[18] and given their similar functions, it is reasonable to assume that YTHDF2 may drive aberrant circRNA translation in GBM. In this study, we showed that circMET is a highly expressed circRNA with intensive m[6]A modification in GBM and encodes MET404 in a YTHDF2-dependent manner. YTHDF2 reportedly mediates methylated RNA degradation, in

cooperation with the translation promoter YTHDF1 in dynamic regulation of protein production[26]. However, YTHDF2 also exerts a disease-specific role by stabilizing MYC and VEGFA transcripts to enhance GBM progression[18]. Here, our data also suggest the distinct function of YTHDF2 in facilitating circRNA translation in a m[6]A-dependent manner in GSCs, which differs from its established role in the RNA decay pathway in other tissues. In contrast, we did not observe an interplay between YTHDF1/3 and circMET translation because MET404 remained unchanged after YTHDF1/3 KD, which may be due to the preferential expression and specific role of YTHDF2 in GSCs. Together, these discoveries indicate that the

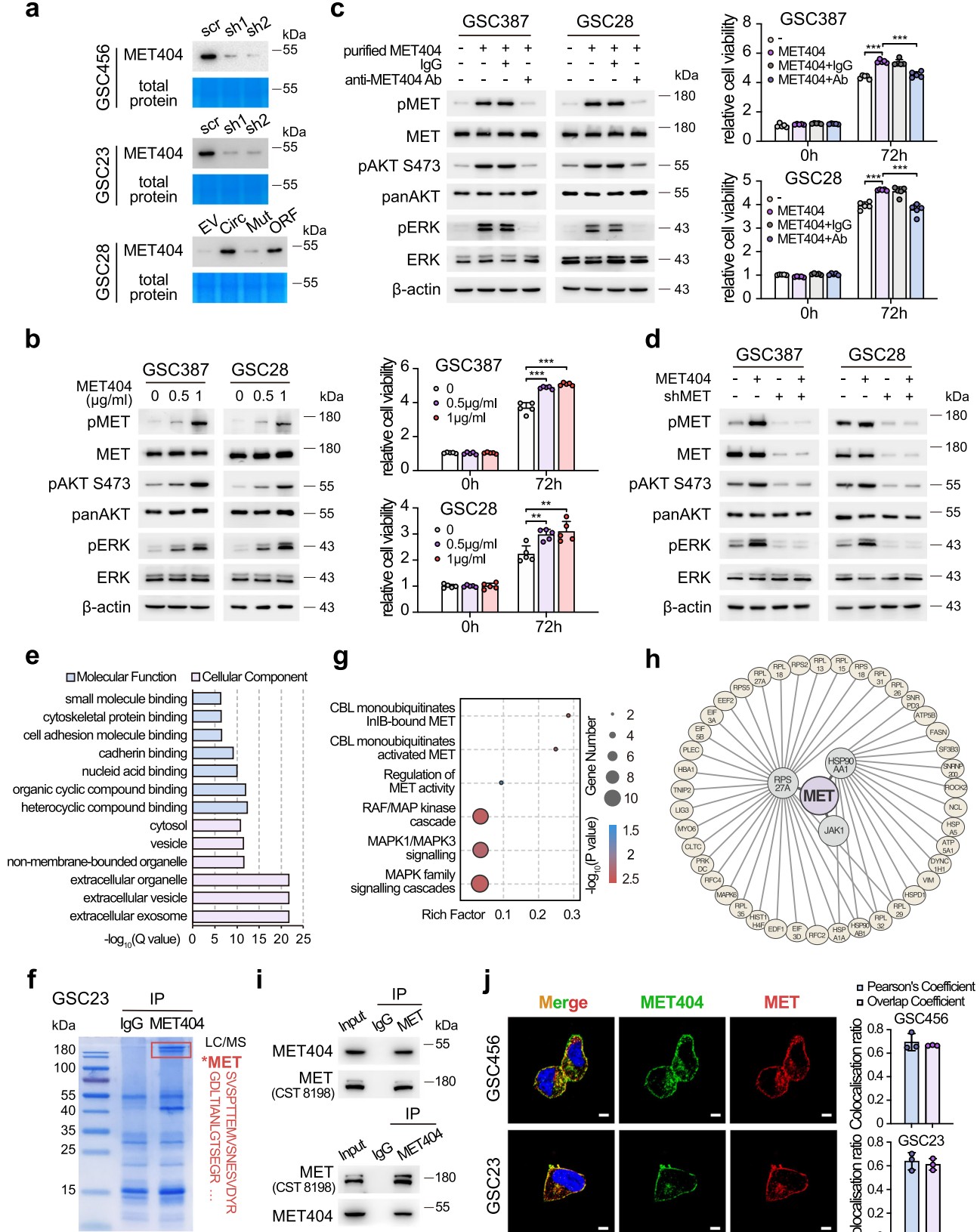

regulation network of m6A-modified RNAs has additional layers of complexity, especially in GBM biology. The underlying causes for the dual function of m6A readers and how the characteristics can be leveraged to potentiate tumour-targeting strategies remain an arena for further investigation.

CircMET was first demonstrated to drive immunosuppression in hepatocellular carcinoma by sponging miRNAs to regulate the Snail/CXCL10 axis[44]. Similar working models have also been reported in lung cancer and renal cell carcinoma[45,46]. Our data demonstrate the critical role of its coding product, MET404, rather than its RNA, in shaping

**Fig. 5 | MET404 is a secreted protein that interacts with MET. a** Immunoblot of concentrated supernatant from the culture medium of GSC456, GSC23 and GSC28 cells with the indicated modifications. EV, empty vector; Circ, stable OE of circMET; Mut, stable OE of circMET with insertion mutation; ORF, stable OE of linearized MET404 ORF vector. Coomassie blue-stained total proteins were used as a loading control. **b** Left, protein levels of phospho-MET and downstream phospho-AKT (S473) and phospho-ERK in GSC387 and GSC28 cells after treatment with purified MET404 at the indicated concentrations. Right, proliferation of GSC387 and GSC28 cells after the indicated treatments. $n = 5$ independent experiments. **c** Left, protein levels of phospho-MET and downstream phospho-AKT (S473) and phospho-ERK in GSC387 and GSC28 cells treated with purified MET404 (1 μg/ml). Specific MET404-neutralizing antibody or IgG control was added as indicated. Right, proliferation of GSC387 and GSC28 cells after the indicated treatments. $n = 5$ independent experiments. 0 vs 0.5 μg/ml, GSC387 $P = 3.34e−05$, GSC28 $P = 0.0022$; 0 vs 1.0 μg/ml, GSC387 $P = 1.09e−05$, GSC28 $P = 0.0049$. **d** Protein levels of phospho-MET and downstream phospho-AKT (S473) and phospho-ERK in GSC387 and GSC28 cells with purified MET404 (1 μg/ml), MET stable KD or both. Ctrl vs MET404, GSC387

$P = 7.05e−07$, GSC28 $P = 1.18e−05$; MET404 vs MET404+Ab, GSC387 $P = 1.86e−05$, GSC28 $P = 5.83e−05$. **e** Gene Ontology analysis of candidate MET404-binding partners. **f** Identification of MET sequences in GSC23 cells immunoprecipitated by an anti-MET antibody using mass spectrometry analysis. **g** Reactome pathway analysis of candidate MET404-binding partners. **h** Protein interaction network with MET as the hub generated using the STRING database based on candidate MET404-binding partners. **i** Whole GSC23 cell lysates were subjected to immunoprecipitation using anti-MET404 and anti-MET antibodies followed by immunoblotting with anti-MET404 and anti-MET antibodies. **j** Left, representative immunofluorescence images of the colocalization of MET404 and MET in GSC456 and GSC23 cells. Scale bar, 5 μm. Right, statistical analysis of GSC456 and GSC23 cells with MET404 and MET colocalization ($n = 3$ independent experiments). The data in (**a**–**d**), (**f**), (**i**) and (**j**) were representative of three independent experiments. The data are presented as the mean ± SD. Unpaired two-tailed Student's $t$ test was used to determine the significance of the differences between the indicated groups where applicable. \*$P < 0.05$; \*\*$P < 0.01$; \*\*\*$P < 0.001$. Source data are provided as a Source data file.

GSC malignancies, as the phenotype upon circMET loss was rescued by putting back MET404 ORF instead of a mutant circMET RNA. In addition, functional peptides encoded by other circRNAs have been proved to be critical players in cancers or other diseases[12–15,47,48]. These findings together indicate complex circRNA functions under different pathological conditions.

In this study, we demonstrate that MET404 serves as a ligand for the MET receptor in addition to HGF, the only previously known MET ligand[49]. MET404 binds a different domain (PSI + IPT1) on the MET receptor that is distant from the HGF-binding site (SEMA), resulting in independent activation of MET signalling and a synergistic effect when combined with HGF. HGF, as the canonical MET ligand, binds MET monomers with different sites and thus forms four interfaces to facilitate a more extended conformation to initiate activation[39]. Here, we validated the capacity of MET404 to induce the dimerization of MET monomers. The current data on the working model of MET404 may lack in-depth structural proof but should be sufficient to support the rationale for targeting MET404 to perturb MET hyperactivation in GBM. The elucidation of the underlying structural basis for activation of the MET receptor by MET404 merits further study and may be used to optimize anti-MET404 strategies.

*MET* is one of the most significantly altered RTKs in GBM in addition to *EGFR* (approximately 4%)[5]. Moreover, MET has been detected in cancer-related endothelial cells, which are critical for neoangiogenesis[50]. To date, *MET* has been proven to be a 'signature gene' in GBM that specifically contributes to the GBM mesenchymal subtype[51]. Although genetic amplification/mutation is relatively rare, the frequency of MET expression in primary GBM varies from 30% to 100%[52]. With its previously identified ligand, HGF (also frequently overexpressed in GBM), HGF/MET signalling is a rational target in GBM treatment. Nevertheless, neither HGF- nor MET-targeted therapy has been found to be effective in current clinical trials[11,53,54]. Similarly, tyrosine kinase inhibitors (TKIs) have been proven to increase RTK mutations that compromise the binding capacity and therapeutic effect during treatment[55]. Here, we identified MET404 as a constitutive ligand for the MET receptor. Together with the fact that MET404 is commonly overexpressed in GBM and highly correlated with the p-MET status, our results suggested that MET404 is an ideal target in MET-hyperactivated GBM, especially when used in combination with other MET inhibitors (Fig. 7h).

Overall, our work identified a molecular target, MET404, which is generated by circMET, in GBM. As a previously unknown MET ligand, MET404 constitutively activates the MET receptor, stimulates downstream effectors independent of HGF and drives the tumorigenesis of GBM. Targeting MET404 with a monoclonal antibody augments the efficacy of anti-HGF/MET therapy, highlighting a translatable strategy for the treatment of pivotal nodes in brain malignancy.

## Methods

### Human glioma and paired adjacent normal tissues
All pathologically diagnosed glioma samples, adjacent normal brain tissues and cerebrospinal fluid used in this study were collected from the Department of Neurosurgery of the First Affiliated Hospital of Sun Yat-sen University with informed consent from the donors. The study was approved by the Ethics Institutional Review Boards of the First Affiliated Hospital of Sun Yat-sen University (No. [2020]322) and complied with all relevant ethical regulations regarding human participants.

### Xenograft studies
Six-week-old female BALB/c-nu mice were purchased from the Laboratory Animal Centre of the First Affiliated Hospital of Sun Yat-sen University. The mice were housed in a temperature-controlled (24–26 °C), humidity-controlled (50–70%) and light-controlled specific pathogen-free animal facility under a 12-h light-dark cycle with free access to food and water. All experimental protocols concerning the handling of mice were approved by the Ethics Institutional Review Boards of the First Affiliated Hospital of Sun Yat-sen University (No. [2021]059).

The mice were injected intracranially with 50,000 of the indicated cells. For the drug treatment experiments, 10 days after the initial tumour implantation, the mice were injected intracranially with IgG (3 μg), anti-MET404 monoclonal antibody (3 μg), onartuzumab (3 μg) or anti-MET404 combined with onartuzumab (3 μg each) through a screw guide (Protech International) every 3 days. Crizotinib (MCE, HY-50878) was successively resolved in dimethyl sulfoxide (DMSO), PEG-300, Tween-80 and saline as recommended by the manufacturer, and administered to the mice via oral gavage (50 mg/kg) every other day, as previously reported[56]. The control mice were administered with the solvent. When the first mouse showed neurologic symptoms that significantly affected their quality of life (such as seizures, ataxia and lethargy and inability to feed), all the mice were euthanized (isoflurane anaesthesia followed by cervical dislocation). Their brains were then harvested, fixed in 4% formaldehyde, embedded in paraffin and then subjected to haematoxylin and eosin staining and IHC staining. For the survival experiments, all the mice were monitored until they showed severe neurologic symptoms or displayed 20% loss of weight and had to be sacrificed humanely. The overall survival curves were calculated with the Kaplan–Meier method and compared by the log-rank test. The maximal tumour burden (~50 mm³) was confirmed not to exceed the ethical limits. The methods have been permitted by the Ethics Institutional Review Boards of the First Affiliated Hospital of Sun Yat-sen University.

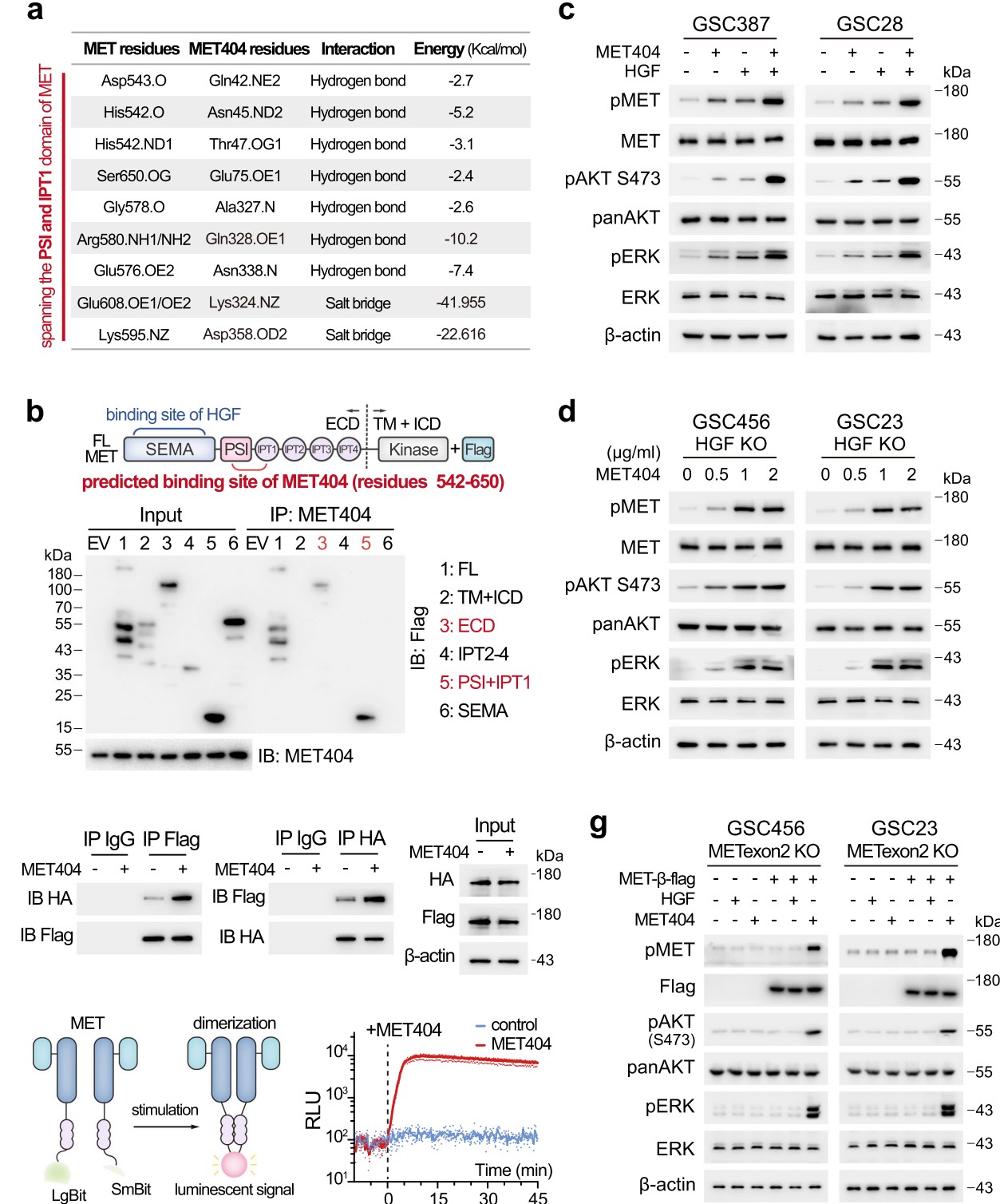

**Fig. 6 | MET404 independently activates the MET receptor. a** List of predicted amino acid contacts between MET404 and MET. **b** Top, illustration of domains on full-length MET. Bottom, 293T cells were transfected with different Flag-tagged MET truncations. Cell lysates were immunoprecipitated with an anti-MET404 antibody and then immunoblotted with an anti-Flag antibody. Representative of three independent experiments. **c** Protein levels of phospho-MET and downstream phospho-AKT (S473) and phospho-ERK in GSC387 and GSC28 cells after 1 h of stimulation with HGF (100 ng/ml), MET404 (1 μg/ml) or both. Representative of three independent experiments. **d** Protein levels of phospho-MET and downstream phospho-AKT (S473) and phospho-ERK in GSC456 and GSC23 HGF KO cells after 1 h gradient stimulation of MET404. Representative of three independent experiments. **e** 293T HGF KO cells were cotransfected with HA- and flag-tagged MET and

subjected to coIP analysis. Cell lysates that were stimulated or not stimulated with MET404 (1 μg/ml) were immunoprecipitated with the anti-HA or anti-flag antibody and then immunoblotted with anti-flag or anti-HA antibody. Representative of three independent experiments. **f** Left, illustration of the NanoBit System. Right, the luminescence of 293T HGF KO cells cotransfected with LgBit- and SmBit-MET was recorded before and after MET404 treatment (1 μg/ml). $n = 3$ independent experiments. **g** HGF (100 ng/ml) or MET404 (1 μg/ml) was used to treat METexon2 KO GSC456/GSC23 cells or those overexpressing the flag-tagged MET β subunit. The protein levels of phospho-MET and downstream phospho-AKT (S473) and phospho-ERK were detected in these cells. Representative of three independent experiments. Source data are provided as a Source data file.

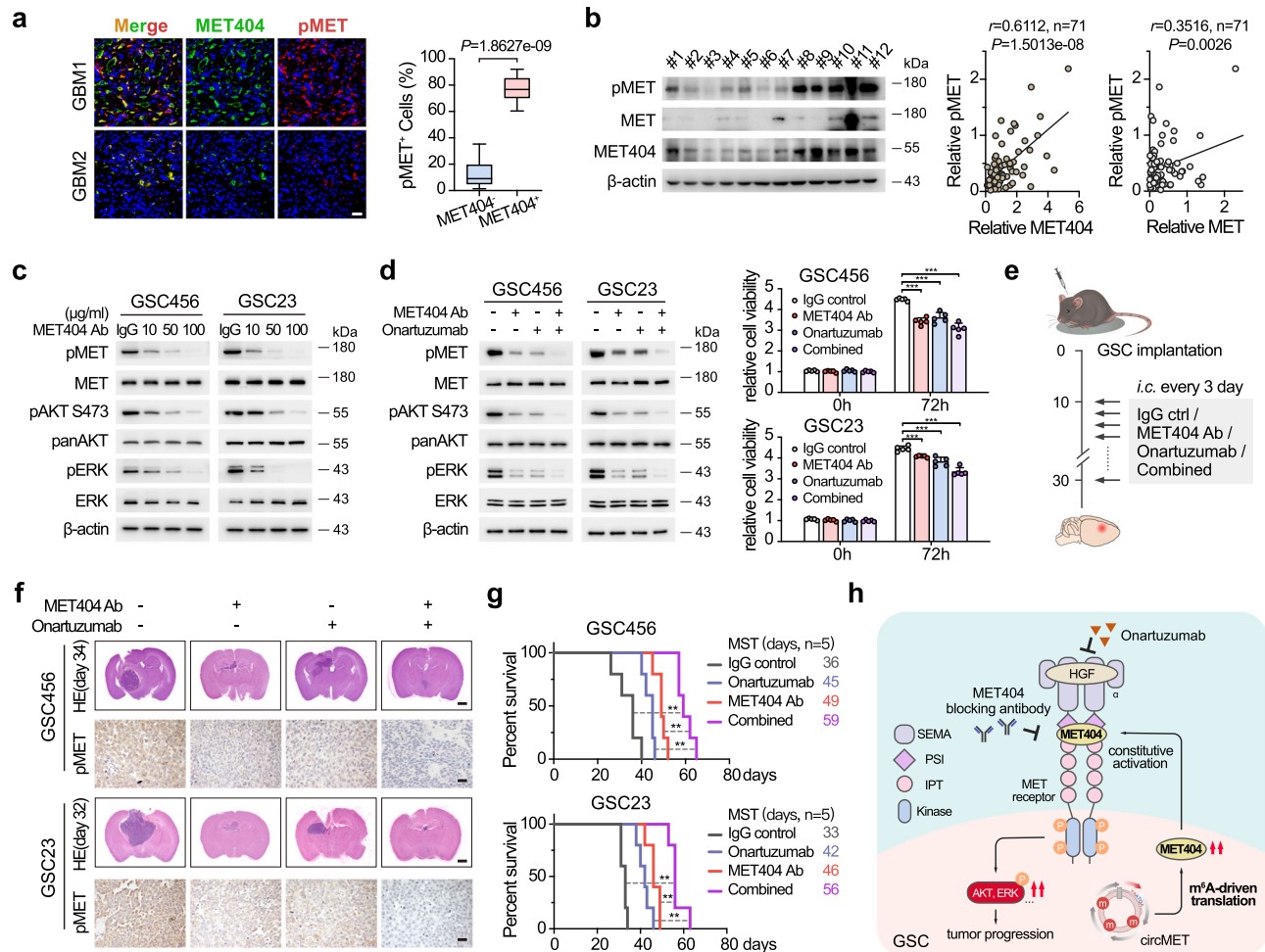

**Fig. 7 | Synergistic effects of targeting MET404 and onartuzumab in restraining GBM progression. a** Left, representative IF images of MET404 and phospho-MET in GBM specimens. Scale bar, 30 μm. Right, proportion of phospho-MET-positive cells among MET404-positive or MET404-negative cells in randomly selected microscopy fields of each IF image (n = 30 independent GBM samples). The data are presented as box plots containing the median (centre line), the first and third quartiles (box limits). The whiskers indicate the maxima and minima. Two-sided Wilcoxon test, P = 1.8627e−09. **b** Left, representative immunoblot of phospho-MET, MET and MET404 in 12 randomly selected independent GBM samples. Right, correlation analysis of MET404 vs. phospho-MET and MET vs. phospho-MET in a cohort of 71 independent GBM samples. r = 0.6112, P = 1.5013e−8 (MET404 vs. phospho-MET); r = 0.3516, two-sided Student's t test, P = 0.0026 (MET vs. phospho-MET). **c** Protein levels of phospho-MET and downstream phospho-AKT (S473) and phospho-ERK in GSC456 and GSC23 cells after treatment with a specific MET404 antibody at the indicated concentrations. Representative of three independent experiments. **d** Left, protein levels of phospho-MET and downstream phospho-AKT (S473) and phospho-ERK in GSC456 and GSC23 cells treated with MET404

antibody, onartuzumab or both. Representative of three independent experiments. Right panel, proliferation of GSC456 and GSC23 cells treated as indicated. n = 5 independent experiments. The data are presented as mean ± SD. Unpaired two-tailed Student's t test. GSC456, IgG vs MET404 Ab, P = 1.09e−06; IgG vs Onar, P = 7.37e−05; IgG vs Combined, P = 3.20e−06. GSC23, IgG vs MET404 Ab, P = 2.94e−04; IgG vs Onar, P = 5.13e−04; IgG vs Combined, P = 6.46e−06. **e** Illustration of the in situ GBM model using nude mice treated with anti-MET404, onartuzumab or combination therapy. i.c., intracranial injection. **f** Representative H&E-stained brain slices and representative IHC images showing the phospho-MET levels in mice implanted with GSC456 or GSC23 and treated with anti-MET404, onartuzumab or both (n = 5 per group). Scale bar, 1 mm (H&E-stained brain slices) or 30 μm (IHC images). **g** Survival analysis of mice subjected to the indicated therapeutic strategy (n = 5 per group). Log-rank test. GSC456, IgG vs Combined, P = 0.0021; MET404 Ab vs Combined, P = 0.0018; Onar vs Combined, P = 0.0021. GSC23, IgG vs Combined, P = 0.0019; MET404 Ab vs Combined, P = 0.0026; Onar vs Combined, P = 0.0018. **h** Summary diagram of the study. Source data are provided as a Source data file.

## Generation of transgenic mice based on CRISPR/Cas9 technology

C57BL/6 mice were used for the generation of transgenic mice. To generate circMET knockout mice, gRNAs targeting the intronic sequence of circMET (gRNA2: AAGACACGGAAGCTATTCTGTGG; gRNA3: TAGGCTTCTCGCTTAAAAGTTGG) and Cas9 mRNA were co-injected into fertilized mouse eggs to generate targeted knockout offspring. To generate MET404 KI (R26-MET404-flag$^{LSL/+}$) mice, the MET404-encoding sequence was cloned into a homologous recombination donor vector and then co-injected with gRNA targeting the Rosa26 locus (gRNA: GGGGACACACTAAGGGAGCTTGG) and Cas9 mRNA into fertilized mouse eggs. P53$^{f/f}$ mice were generated using a

standard KI approach in which P53 exons 5–7 were flanked by loxP sites. GFAP-Cre mice, P53$^{f/f}$ mice and R26-MET404-flag$^{LSL/+}$ mice were hybridized to generate control (P53$^{f/f}$; R26-MET404-flag$^{+/+}$; GFAP-Cre) mice and MET404 KI (P53$^{f/f}$; R26-MET404-flag$^{LSL/+}$; GFAP-Cre) mice. Gene-targeted mice were identified by PCR screening and DNA sequencing. The mice were monitored daily, and euthanized (isoflurane anaesthesia followed by cervical dislocation) when they displayed severe neurological symptoms, 20% loss of weight or became moribund. The maximal tumour burden (~50 mm³) was confirmed not to exceed the ethical limits. All experimental protocols concerning the handling of mice were approved by the Ethics Institutional Review Boards of the First Affiliated Hospital of Sun Yat-sen University (No. [2021]059).

## Acute liver injury model

This model was established closely following previously reported methods[36,57]. In brief, 8- to 12-week-old male wild-type and circMET KO mice (C57BL/6 background) received one intraperitoneal injection of 1 mL/kg CCl$_4$ (Macklin #56-23-5, diluted in mineral oil (Sigma, #M5310) at a ratio of 1 to 4). The uninjured controls was administered 1 mL/kg filtered mineral oil. After 6 and 24 h, the liver was collected for further immunoblot analysis. On days 1, 2, 3, 5 and 7, the liver tissue was collected and fixed in 4% paraformaldehyde (PFA), embedded in paraffin and stained with haematoxylin and eosin.

## Cell culture studies

GSCs, including 387, 456, 28 and 23, were kindly provided by Dr. Jeremy Rich, UPMC. Background information on these GSCs, including genetic alterations, transcriptional subtype and RTK expression levels, is provided in Supplementary Fig. 9. These cells were cultured in DMEM/F12 medium (Gibco) supplemented with B27 supplement (Life Technologies), bFGF and EGF (20 ng m l$^{-1}$ each, R&D systems). 293T (CRL-11268) and U118 (HTB-15) cells were purchased from ATCC. U251 and SNB19 cells were kindly provided by Dr. Suyun Huang, VCU. These cell lines were cultured in Dulbecco's modified Eagle's medium (Gibco) supplemented with 10% foetal bovine serum (Gibco) according to standard protocols. NHA cells were purchased from Lonza and cultured using an AGM$^{TM}$ Astrocyte Growth Medium Bullet Kit$^{TM}$ (Lonza), as recommended by the manufacturer. Primary human NSCs were obtained from Thermo Fisher (A15654) and Gibco (TM A15654), and cultured with StemPro NSC SFM (A10509-01) supplemented with 2 mM GlutaMAX-I supplement (35050), 6 U ml$^{-1}$ heparin (Sigma, H3149) and 200 µM ascorbic acid (Sigma, A8960). All cells used in this study were cultured in a humidified incubator at 37 °C with 5% CO$_2$ and were free of mycoplasma contamination (regularly tested). The cell lines were authenticated using short tandem repeat (STR) fingerprinting method. In experiments concerning MET downstream signalling, GSCs were maintained in plain DMEM/F12 medium (free of EGF and bFGF) for 24 h before stimulation or other treatment.

## Constructs and reagents

The pEZ-ALKBH5 WT and pEZ-ALKBH5 mutant (H204A) plasmids were kind gifts from Dr. Shirong Cai (First Affiliated Hospital of Sun Yat-sen University, China). The YTHDF2 WT, W432A mutant and W486A mutant plasmids were kindly provided by Prof. Xiuxing Wang (Nanjing Medical University, China). The circMET overexpression plasmid, circMET Mut and circMET ORF plasmids, and His-MET404 plasmid were generated by chemical gene synthesis, and the pCDH-CMV-MCS-EF1-copGFP-T2A-Puro vector was used as the plasmid backbone (Generay Biotech, China). The sequences of the circMET overexpression plasmids are given in Supplementary Table 2. Flag-tagged full-length and truncated MET overexpression plasmids were generated by chemical gene synthesis and inserted into the pcDNA 3.1(+) construct (Generay Biotech, China). All the siRNAs and shRNAs used in this study were generated by GenePharma or WZ Biosciences Inc., China. The target sequences are listed in Supplementary Table 3. Plasmids and siRNAs were transfected using Lipofectamine 3000 (Invitrogen) according to the manufacturer's protocol. Recombinant human HGF (Z03229) was purchased from GenScript, China. Onartuzumab (ATAD00251) was generated by AtaGenix, China. MET404 was purified from His-MET404 transfected 293T cells using the BeaverBeads IDA-Nickel Kit (7050-K10).

## Lentivirus production and stable cell line construction

Vectors expressing circMET, circMET Mut, circMET ORF, His-MET404 (pCDH-CMV-MCS-EF1-copGFP-T2A-Puro as plasmid backbone) were cotransfected with the packaging vectors psPAX2 (Addgene) and pMD2.G (Addgene) into 293T cells for lentivirus production using Lipofectamine 3000 (Invitrogen) following the manufacturer's protocol. GSCs were infected with the above lentiviruses using polybrene (8 µg/ml, GenePharma) for up to 48 h. Starting 72 h after infection, the cells were screened with 2 µg/ml puromycin for 7 days.

## N$^6$-methyladenosine immunoprecipitation (m$^6$A IP) and sequencing

After total RNA was extracted, the enriched RNA was broken into short fragments (~100 nt) using fragmentation buffer. The RNA was divided into two parts, and one part was used as the input (no IP experiment was performed). The other part was enriched with a m$^6$A-specific antibody. The input and enriched RNA were reverse transcribed into cDNA with random primers. Next, the cDNA fragments were end repaired and ligated to Illumina sequencing adaptors. The quality-tested library was sequenced using an Illumina NovaSeq$^{TM}$ 6000 by Gene Denovo Biotechnology Co. (Guangzhou, China).

The raw reads obtained from the sequencing machines were processed using fastp[58] (version: 0.20.0) software to obtain high-quality clean reads. The short read alignment tool Bowtie2[20] (version 2.2.8) was used for mapping the reads to the ribosome RNA (rRNA) database. The rRNA mapped reads were removed. The remaining reads were further used for alignment and analysis. The rRNA-removed reads from each sample were then mapped to the reference genome by HISAT2[21] (version 2.1.1). After alignment with the reference genome, the reads that could be mapped to the genomes were discarded, and the unmapped reads were then collected for circRNA identification. Anchor reads that aligned in the reverse orientation (head-to-tail) indicated circRNA splicing and were subjected to find_circ[22] (version 1.2) and CIRIquant[24] (version 2.1.1) to identify circRNAs. The anchor alignments were then extended such that the complete reads aligned and the breakpoints were flanked by GU/AG splice sites. A candidate circRNA was called if it was supported by at least two unique back-spliced reads in at least one sample. CircRNAs were blasted against circBase[59] for annotation. Those that could not be annotated were defined as novel circRNAs. The source gene is the origin gene of a circRNA. We performed KEGG[60] enrichment analysis of source genes to study the main functions of these source genes of circRNAs.

## Ribosome-nascent chain complex (RNC) sequencing and bioinformatics analysis

RNC-seq was performed as described previously[23]. Briefly, cells were pretreated with 100 µg/ml cycloheximide for 15 min before lysis. Cell lysates were transferred onto the surface of 30% sucrose RB buffer and then subjected to ultracentrifugation at 185,000 × g for 5 h at 4 °C for RNC precipitation. RNA in total cell lysates and RNC-mRNAs were extracted and subjected to RNA sequencing. RNC-RNA was extracted using a TRIzol reagent kit (Invitrogen, Carlsbad, CA, USA) according to the manufacturer's protocol. The RNA quality was assessed with an Agilent 2100 Bioanalyzer (Agilent Technologies, Palo Alto, CA, USA) and checked using RNase-free agarose gel electrophoresis. After RNC-RNA was extraction, rRNAs were removed to retain mRNAs and ncRNAs. The enriched mRNAs and ncRNAs were broken into short fragments using fragmentation buffer and reverse transcribed into cDNA with random primers. Second-strand cDNA was synthesized with DNA polymerase I, RNase H, dNTPs (dUTP instead of dTTP) and buffer. Next, the cDNA fragments were purified with a QiaQuick PCR extraction kit (Qiagen, Venlo, The Netherlands), end repaired, poly(A) conjugated, and ligated to Illumina sequencing adaptors. Then, uracil-N-glycosylase (UNG) was used to digest the second-strand cDNA. The digested products were size selected by agarose gel electrophoresis, PCR amplified, and sequenced using an Illumina NovaSeq$^{TM}$ 6000 by Gene Denovo Biotechnology Co. (Guangzhou, China).

The method of circRNA detection for RNC-seq data was the same as that used for m$^6$A-seq. To quantify circRNAs, back-spliced junction

reads were scaled to reads per million mapped reads (RPM). To identify differentially expressed circRNAs across samples or groups, the edgeR package[61] (version 3.12.1) [http://www.r-project.org/] was used. We identified circRNAs with a fold change ≥2 and a P value ≤ 0.05.

## N[6]-methyladenosine immunoprecipitation (m[6]A IP) and RT−qPCR analysis

M[6]A IP was conducted using a Methylated RNA Immunoprecipitation Kit (BersinBio, Bes5203) according to the manufacturer's instructions. Briefly, $1 \times 10^7$ cells were prepared and subjected to RNA extraction using TRIzol Reagent (Invitrogen). Immunoprecipitation was performed using an anti-m[6]A antibody (Synaptic Systems, 202003) and protein A/G beads. RNA from the eluate (m[6]A positive) and supernatant (m[6]A negative) was purified separately and subjected to RT−qPCR analysis.

## RNA-binding protein immunoprecipitation (RIP) assay and sequencing

The RIP assay was performed as previously described, with minor modifications[62]. Briefly, 5 μg anti-YTHDF2 antibody (Proteintech, 24744-1-AP) was precoated onto protein A/G beads at 4 °C overnight. The cell pellet was lysed using ice-cold polysome extraction buffer supplemented with RNase inhibitors and protease inhibitors. For each RIP, 1000 μg lysate was incubated with precoated beads for 1.5 h at 4 °C with rotation. After incubation, the beads were washed 5 times and subjected to DNA and protein digestion and RNA extraction. The isolated RNA was then subjected to sequencing and RT-qPCR analysis. For sequencing analysis, the input and YTHDF2-antibody-enriched RNA was broken into short fragments (approximately 250 nt) using fragmentation buffer and then reverse-transcribed into cDNA with random primers. Next, the cDNA fragments were end repaired and ligated to Illumina sequencing adaptors. The quality-tested library was sequenced using an Illumina NovaSeq 6000 by Gene Denovo Biotechnology Co. (Guangzhou, China). The method for circRNA detection used for the YTHDF2 RIP-seq data was the same as that used for the m[6]A-seq data.

## RT-qPCR analysis

PrimeScript™ RT Master Mix (Takara) was used for RNA reverse transcription according to the manufacturer's instructions unless otherwise indicated. Quantitative polymerase chain reaction (qPCR) was performed using TB Green® Premix Ex Taq™ II (Tli RNaseH Plus) (Takara). The primer sequences are listed in Supplementary Table 4. The relative expression levels were calculated according to $2^{-\Delta\Delta CT}$.

## RNase-R treatment

Total RNA was extracted and then treated with RNase-R (Lucigen) at 37 °C for 15 min. The RNase-R resistance of circMET was evaluated by RT-qPCR and Northern blot analysis.

## Actinomycin D assay

First, 293T cells were treated with actinomycin D (2 μg/ml, HY-17559, MedChem Express) for 0, 4, 8, 12 and 24 h. The cells were then harvested at the indicated time points, and the relative RNA levels of circMET and linear MET were analysed by RT-qPCR and normalized to the values obtained for the 0 h group.

## RNA fluorescence in situ hybridization (FISH)

CY3-labelled oligonucleotide probes complementary to the circMET junction region were designed using the Clone Manager suite of analysis tools (Sci Ed Central, listed in Supplementary Table 3). The indicated cells were precoated with poly-L-ornithine and laminin (Sigma) on a cover glass-bottom confocal dish and cultured overnight. FISH assays were performed using an RNA FISH kit (GenePharma, China) according to the manufacturer's instructions. Nuclei were stained with 4,6-diamidino-2-phenylindole (DAPI). Images were acquired using a ZEISS LSM 880 with Airyscan.

## RNA subcellular isolation

Cytoplasmic and nuclear fractions were isolated using the reagents supplied with the RNA subcellular isolation kit (Active Motif). Briefly, cells were lysed in complete lysis buffer and incubated for 10 min on ice. After centrifugation, the supernatant was transferred for cytoplasmic RNA extraction, and the remaining pellet was collected for nuclear RNA purification. The RNA products were subjected to RT-qPCR analysis.

## Northern blot

Northern blotting was conducted using NorthernMax-Gly Kit (Invitrogen, AM1946), and a Chemiluminescent Biotin-label Detection Kit (Beyotime, D3308) following manufacturer's protocols. The circMET junction- and exon-specific probes were listed in Supplementary Table 3. The extracted RNA was separated by agarose gel electrophoresis and transferred to nylon membrane using capillary transfer. After the membrane was fixed by UV, specific probes were added and washed using reagents provided with the kit. Chemiluminescent signals were recorded by Amersham ImageQuant 800 and analysed using Image Lab 6.0.1.

## Polysome profiling assay

The details of the polysome profiling analysis were described previously[63]. Briefly, $1 \times 10^7$ cells were treated with 100 μg/mL cycloheximide in DMSO for 5 min at 37 °C and then harvested for polysome profiling. The cells were lysed in 500 μl of polysome lysis buffer and then centrifuged. The supernatant was then loaded onto a 5–50% (w/v) sucrose density gradient, ultracentrifuged at $20,000 \times g$ for 2 h at 4 °C in a Beckman SW41 rotor and subsequently fractionated using a Bio-Comp PGFip Piston Gradient Fractionator Model 152. The absorbance at 254 nm was measured using an absorbance detector connected to the fraction collector. RNA was extracted from different fractions, and RT-qPCR was conducted to evaluate the distribution of targets.

## Immunoblotting

Briefly, after extraction with RIPA buffer supplemented with protease inhibitor and phosphatase inhibitor cocktails (MedChem Express) and quantification with a BCA kit (Thermo Fisher), equal amounts of protein from cell lysates or tissue lysates were denatured by boiling, resolved by SDS−polyacrylamide gels and transferred to polyvinylidene fluoride (PVDF) membranes. After blocking with 5% non-fat milk, the membranes were consecutively incubated with the indicated primary antibodies and HRP-conjugated secondary antibodies (SeraCare, 5220-0336/5220-0341, 1:10,000). The chemiluminescence signals were detected using Clarity™ Western ECL Substrate (Bio-Rad). The anti-MET404 mouse monoclonal antibody was generated by GenScript, China (clone 1C11, 1:1000). The other primary antibodies are the following: anti-ALKBH5 (Proteintech, 16837-1-AP, 1:2000); anti-YTHDF2 (Proteintech, 24744-1-AP, 1:2000); anti-YTHDF1 (Proteintech, 17479-1-AP, 1:1000); anti-YTHDF3 (Proteintech, 25537-1-AP, 1:2000); anti-P53 (Proteintech, 60283-2-Ig, clone 6C4B6, 1:5000); anti-MET (N-terminus specific, Abcam, ab51067, clone EP1454Y, 1:1000); anti-MET (Cell Signaling Technology, 8198, clone D1C2, 1:1000), anti-phospho-MET Tyr1234/1235 (Cell Signaling Technology, 3077, clone D26, 1:1000); anti-AKT (Cell Signaling Technology, 4691, clone C67E7, 1:1000); anti-phospho-AKT Ser473 (Cell Signaling Technology, 4060, clone D9E, 1:1000), anti-ERK1/2 (SAB, 29935, 1:1000), anti-phospho-ERK1/2 (SAB, 52094, clone S05-2H9, 1:1000), anti-EGFR (Boster, A00023-4, 1:1000), anti-phospho EGFR Y1068 (Abcam, ab40815, clone EP774Y, 1:1000), anti-PDGFR alpha (Abcam, ab248689, clone EPR5480,

1:1000), anti-phospho PDGFR alpha Y720 (Abcam, ab134068, clone EP2478, 1:1000), anti-FGFR1 (Abcam, ab76464, clone EPR806Y, 1:500), anti-phospho FGFR1 Y654 (Abcam, ab59194, 1:1000), anti-panTrk (Abcam, ab76291, clone EP1058Y, 1:1000), anti-phospho TrkB Y705 (Abcam, ab229908, clone EPR22298-67, 1:1000), anti-METTL3 (Proteintech, 15073-1-AP, 1:1000), anti-6x-His tag (Abcam, 18184, clone HIS.H8, 1:1000), anti-flag tag (F1804, Sigma-Aldrich, clone M2, 1:5000), anti-HA tag (Sigma-Aldrich, H6908, 1:1000), anti-β-actin (Sigma-Aldrich, A1978, clone AC-15, 1:1000) and anti-β-tubulin (Cell Signaling Technology, 2128, clone 9F3, 1:1000).

### Processing of human glioma tissue and CD133$^+$ cell isolation

Fresh human glioma tissue was minced, digested (collagenase IV (1 mg/ml, Gibco), DNase I (20 U/ml, Sigma-Aldrich), hyaluronidase (0.01%, Solarbio) and DMEM/F12 (Gibco)) for 40 min at 37 °C, and sequentially filtered through 70-μm and 40-μm strainers. Myelin was removed by Debris Removal Solution (130-109-398, Miltenyi Biotec) according to the manufacturer's instructions. Red blood cells were then lysed with RBC lysis buffer (C3702, Beyotime). The collected cell pellets were then resuspended in FACS staining buffer (PBS containing 2% FBS) and stained with anti-CD133 PE (566593, BD Biosciences, clone W6B3C1, 1:50). Homologous IgG was used as an isotype control antibody. Fixable Viability Dye eFluor 520 (65-0867-14, eBioscience) was applied to rule out dead cells. Live CD133-positive and CD133-negative cells were sorted with a BD FACSAria II Cell Sorter. The collected cells were then subjected to RNA or protein extraction before qPCR or immunoblot analysis.

### LC−MS analysis

Proteins were separated via SDS-PAGE and subjected to digestion. Chymotrypsin was used to analyse the specific tail of MET404. Otherwise, trypsin was used for digestion. The digested peptides were analysed with a QExactive mass spectrometer (Thermo Fisher). The fragment spectra were analysed using the National Centre for Biotechnology Information nonredundant protein database with Mascot. For LC/MS analysis of membrane proteins, a Membrane and Cytosol Protein Extraction Kit (Beyotime, P0033) was used to extract membrane proteins from the indicated cells. GO enrichment (http://www.geneontology.org/) and Reactome enrichment analysis (https://reactome.org/) of the proteins detected from Met404 Co-IP were conducted. The protein interaction network was generated using the STRING database (https://cn.string-db.org/). The network around MET was plotted using Cytoscape[64].

### Immunoprecipitation (IP)

Cells were lysed in co-IP soft RIPA Lysis Buffer (Beyotime, China) supplemented with protease and phosphatase inhibitors. The supernatant was collected and subjected to immunoprecipitation using the indicated primary antibodies at 4 °C overnight. Then, the lysates were then incubated with 30 μL of protein A/G agarose (Gibco) for 2 h at room temperature. The collected agarose-protein complexes were centrifuged and washed with cold PBST (PBS containing 0.1% Tween 20) for five times and then subjected to SDS-PAGE and analysed by LC−MS or immunoblotting.

### Immunofluorescence (IF) staining

Seeded cells on coverslips or frozen glioblastoma sections were fixed with 4% paraformaldehyde for 15 min, permeabilized with PBS containing 0.1% Triton X-100 for 5 min at room temperature, blocked with 5% BSA in PBS, and then incubated with an anti-MET404 monoclonal antibody (GenScript clone 1C11, 1:200), anti-MET antibody (Abcam, ab51067, clone EP1454Y, 1:200), and anti-phospho-MET Tyr1234/1235 (Cell Signaling Technology, 3077, clone D26, 1:200) overnight at 4 °C and then with the appropriate secondary fluorescently labelled antibodies (Invitrogen, A-21202/A-21203/A-21206/A-21207, 1:500) for 1 h at

room temperature. Nuclei were counterstained with DAPI. Images were acquired using a ZEISS LSM 880 with Airyscan.

### Immunohistochemistry (IHC)

Paraffin-embedded brain tissues were sectioned, and a gradient of concentrations of xylene and ethanol was used for dewaxing and hydration. Antigen retrieval was performed using a microwave for 20 min in 0.01 M citrate buffer (pH 6.0). After blocking with 3% $H_2O_2$ and then 10% FBS, the samples were incubated with primary antibody overnight at 4 °C and secondary antibody for 30 min at room temperature. Immunodetection was performed using DAB solution. The tissues were counterstained with haematoxylin. The primary antibodies used in this study are the following: anti-MET404 monoclonal antibody (GenScript clone 1C11, 1:200); anti-YTHDF2 (Proteintech, 24744-1-AP, 1:200); anti-GFAP (Abcam, ab7260, 1:500); anti-Nestin (Abcam, ab221660, clone EPR22023, 1:1000); anti-Ki67 (Abcam, ab15580, 1:200); and anti-phospho-MET (Cell Signaling Technology, 3077, clone D26, 1:200).

### Enzyme-linked immunosorbent assay (ELISA)

MET404 capture antibody was diluted to an appropriate concentration with ELISA coating buffer (Thermo Scientific, 28382). A 96-well MaxiSorp plate (Thermo Scientific, 468667) was coated with the diluted antibody and incubated overnight at 4 °C. The plate was then washed (Pierce, 28341) and blocked (Thermo Scientific, DS98200). Purified MET404 protein and CSF samples were prepared with appropriate concentrations in the blocking buffer, added to the designated wells and incubated for 2 h at room temperature with gentle shaking. The wells were washed and incubated with diluted MET404 detection antibody for another hour at room temperature. After the detection antibody was washed away, HRP-conjugated secondary antibody was added. After the last wash of the secondary antibody, a working solution of chemiluminescent substrate (Thermo Scientific, 37070) was added, and the plate was incubated. The absorbance was measured at 450 nm and analysed using SkanIt 6.1.

### NanoBit System

The NanoBit Protein: Protein interactions system was obtained from Promega (N2014) and used according to the manufacturer's protocol. Briefly, full-length MET was constructed into the provided vector to fuse LgBit and SmBit onto its C-terminus. The constructs were then transiently transfected into HEK293T cells using Lipofectamine 3000 (Invitrogen) following the manufacturer's protocol. After 24 h, the cells were collected and seeded into 96-well plates with reduced serum medium (Thermo Fisher, 31985070). Before stimulation, Nano-Glo Live Cell Reagent provided with the kit was added and the baseline luminescence was measured using a Multimode Microplate Reader (Thermo Scientific Varioskan LUX) for 10 min. After that, purified MET404 protein (1 μg/ml) or an equivalent amount of Opti-MEM was added to each well, and the plate was gently mixed. The luminescence was measured continuously for up to 45 min. The data were recorded using SkanIt 6.1 and plotted using GraphPad Prism 8.0.

### Molecular docking

Protein−protein docking in the ClusPro server[65] was used for molecular docking simulations of complexes of MET with MET404. The crystal structure of MET was downloaded from the RCSB Protein Data Bank (PDB ID: 2UZY). For protein docking, the smaller protein (a smaller number of residues) is usually set as the ligand and the other is considered as the receptor. The ligand was subjected to 70,000 rotations. For each rotation, the ligand was translated along the x, y, and z axes relative to the receptor on a grid. One translation with the best score was chosen from each rotation. Of the 70,000 rotations, the 1000 rotation/translation combinations with the lowest score were selected.

Subsequently, greedy clustering of these 1000 ligand positions with a 9 Å C-alpha RMSD radius was performed to find the ligand positions with the most "neighbours" of 9 Å, i.e., cluster centres. The top ten cluster centres with the most cluster members were then retrieved, and the intermolecular contacts from the most likely pose were further evaluated.

## Proliferation assay

Cell proliferation experiments were conducted by seeding cells of interest at a density of 2000 cells per well into 96-well plates. At the indicated time points, cell viability was determined using a Cell Counting Kit-8 (Dojindo). All data were normalized to the data at day 1 and presented as the mean ± SD. All experiments were performed in triplicate.

## Neurosphere formation assay

Briefly, decreasing numbers of cells per well (20, 10, 5, 2 and 1) were plated into 96-well plates. The presence of neurospheres in each well was recorded seven days after plating. Extreme limiting dilution analysis was performed using software available online (http://bioinf.wehi.edu.au/software/elda). All experiments were performed in triplicate.

## EdU incorporation assay

The indicated cells were seeded on precoated coverslips, and EdU incorporation rates were determined using a BeyoClick™ EdU Cell Proliferation Kit with Alexa Fluor 594 (Beyotime, China) according to the manufacturer's instructions. All experiments were performed in triplicate.

## Statistical analysis

Statistical tests were conducted using GraphPad Prism (Version 8.0) software and Microsoft Excel (Version 16.74) unless otherwise indicated. The data are presented as the means ± standard deviations (SD) from three independent experiments. For the comparison of parametric data between glioma samples and adjacent normal brain tissues, paired two-tailed Student's t tests were used. For other parametric data, unpaired two-tailed Student's t tests or ANOVA tests were used. Overall survival curves were assessed with the Kaplan–Meier method and compared by the Log-rank test. The correlations were calculated by Pearson correlation analysis. A level of $P < 0.05$ was used as the cut-off for significant differences. For each experiment, the data are representative of three replicates, and similar results were obtained.

## Reporting summary

Further information on research design is available in the Nature Portfolio Reporting Summary linked to this article.

## Data availability

The m⁶A-seq data of human GBM samples have been deposited in the National Genomics Data Centre (GSA-human database) under the accession code HRA002365. The RNC-seq and YTHDF2 RIP-seq data have been deposited in the NCBI database under the accession ID PRJNA973050. The remaining data are available within the Article, Supplementary Information and Source data file. Source data are provided with this paper.

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

## Acknowledgements

We gratefully thank all patients who participated in these studies. This work was supported by the National Natural Science Distinguished Youth Foundation of China (82125024 to N.Z.), Special Funds of the National Natural Science Foundation of China (82341007 to N.Z.), Guangdong Province Regional Joint Fund - Key Project (2022B1515120023 to N.Z.), the Major Program of the National Natural Science Foundation of China (82192891 to Y.S., 82192894 to N.Z.), the National Natural Science Outstanding Youth Foundation of China (81922056 to Y.S.), the National Natural Science Foundation of China (82072779 to X.W.), and the Science and Technology Planning Project of Guangzhou (202103000019 to N.Z.).

## Author contributions

Y.S., Xiuxing W., F.B. and N.Z. designed the experiments. J.Z., Xujia W., Y.G. and J.C. performed the experiments. J.Z., Xujia W., M.Z., H.Z., H.Q., Y.S. and N.Z. analysed the data. J.Z., Xujia W. and N.Z. wrote the manuscript. Y.S., Xiuxing W., F.B., J.Y., X.Y., N.H. and F.X. provided scientific input and helped edit the manuscript.

## Competing interests

The authors declare no competing interests.

## Additional information

[1]Department of Neurosurgery, The First Affiliated Hospital of Sun Yat-sen University, Guangzhou, Guangdong 510080, China. [2]Guangdong Provincial Key Laboratory of Brain Function and Disease, Guangdong Translational Medicine Innovation Platform, Guangzhou, Guangdong 510080, China. [3]Department of Scientific Research Section, The First Affiliated Hospital of Sun Yat-sen University, Guangzhou, Guangdong 510080, China. [4]Institute of Pathology and Southwest Cancer Centre, Southwest Hospital, Third Military Medical University (Army Medical University), Chongqing 400038, China. [5]Key Laboratory of Tumour Immunopathology of the Ministry of Education of China, Southwest Hospital, Third Military Medical University (Army Medical University), Chongqing 400038, China. [6]National Health Commission Key Laboratory of Antibody Techniques, School of Basic Medical Sciences, Nanjing Medical University, Nanjing, Jiangsu 211166, China. [7]Department of Cell Biology, School of Basic Medical Sciences, Nanjing Medical University, Nanjing, Jiangsu 211166, China. [8]Jiangsu Provincial Key Laboratory of Human Functional Genomics, School of Basic Medical Sciences, Nanjing Medical University, Nanjing, Jiangsu 211166, China. [9]Institute for Brain Tumors, Jiangsu Key Lab of Cancer Biomarkers, Prevention and Treatment, Collaborative Innovation Center for Personalized Cancer Medicine, Nanjing Medical University, Nanjing, Jiangsu 211166, China. [10]Biomedical Pioneering Innovation Center (BIOPIC), School of Life Sciences, Peking University (PKU), Beijing, China. [11]Beijing Advanced Innovation Center for Genomics (ICG), Peking University, Beijing, China. [12]These authors contributed equally: Jian Zhong, Xujia Wu, Yixin Gao, Junju Chen. ✉e-mail: drxiuxingwang@163.com; fbai@pku.edu.cn; drshiyu@126.com; zhangnu2@mail.sysu.edu.cn

