## [Peer Review File · Nature Communications]

Circular RNA encoded MET variant is a targetable factor in glioblastomaEditorial Note: Parts of this Peer Review File have been redacted as indicated to remove third-party material where no permission to publish could be obtained.

Reviewers' comments:

Reviewer #1 (Remarks to the Author):

This manuscript claims the discovery of a new MET protein variant (MET 404) that is encoded by circular MET RNA (circMET) and that is facilitated by the N6 methyladenosine reader YTHDF2. It shows that MET404 promotes glioblastoma (GBM) tumorigenesis by interacting with the MET receptor and promoting MET signaling. Targeting MET404 with a newly developed antibody synergizes with the MET monoclonal antibody onartuzumab in inhibiting in vivo GBM xenograft growth.

This appears to be an interesting discovery. However, the experimental plan is severely lacking in rigor and specificity, making the interpretation of the data and the conclusions very questionable. In general, the approaches cannot distinguish between MET404 and MET transcripts and protein as well as previously shown non-coding effects of circMET. Also, the origin of MET404 from circMET is not convincingly established. Specifically:

- The shRNA used to knock down (KD) circMET is not specific to MET404. It was not tested against linear MET mRNA transcripts. Worse, a quick BLAST alignment by this reviewer revealed that one siRNA aligns with MET mRNA and the other with several other oncogenes. The KD effects that are attributed to MET404 in several data in the manuscript can therefore very well be the resultant of actual MET or other oncogene KD.
- Both forward and reverse circMET primers fully align with MET mRNA, creating the same issue described above.
- The antibody that was generated to detect (in several experiments), immunoprecipitate and inhibit MET404 was not shown to be specific for MET404. In fact, supplementary figure 2e and 2g suggest that it actually detects the full MET protein. The interpretation of all data generated using this antibody is therefore questionable.
- CircMET has been discovered and published before and shown to have oncogenic effects that are dependent on its not non-coding effects. Even if the circMET KD effects were specific, the functional consequences could be due to factors other than MET404. The previous publications on circMET are not even cited.
- The demonstration that MET404 is not MET linear splicing variant is not very convincing.

Reviewer #3 (Remarks to the Author):

The authors show that a circular RNA from the MET gene is an important player in glioblastoma. It encodes a novel MET variant protein (MET404) that can form a constitutively activated MET receptor whose activity does not require HFG stimulation. The data are intriguing, but I have a number of questions especially with regards to stoichiometry. Some key experimental details are also lacking that makes it difficult to judge the relevance of the conclusions.

(1) Figure 1: The authors have used a single annotation algorithm, `find_circ`, to annotate circular RNAs but it is well established that these algorithms can have high false positive rates. At least one additional annotation algorithm needs to be used, as is common in the field. For non-experts in the circRNA field, it would be helpful to include an additional few sentences on p.4-5 to clarify how circRNAs can be identified using the backsplicing junction sequences.

(2) There is no attention to stoichiometry in this paper. How does the level of circMET compare to the linear MET mRNA? How does the level of the MET404 protein compare to the standard MET protein? The protein data are particularly important given the authors' model for how MET404 functions. The authors should, for example, use an antibody that recognizes both protein isoforms. It is also not clear how the levels of MET404 compare to the MET receptor, and if it is sufficiently

high to justify the hyperactivated MET signaling observed in GBM.

(3) Northern blots are a standard approach for proving/validating high circular RNA expression, but these are lacking in the manuscript.

(4) Figure 2I, J: Is the standard MET protein also over-expressed in these particular cancerous tissues?

(5) Figure 3A: The authors have over-expressed circMET, but no validation is provided to show that the circRNA was over-expressed or how cleanly it was done, e.g. are linear RNAs also over-expressed. This is best done using Northern blots. There are also no details explaining how the over-expression approach works. Sufficient details need to be provided in the methods so that the work can be replicated by a reader if they desired to do so.

(6) Fig 6: Addition of MET404 protein is used, but it is unclear if physiologically relevant levels of purified protein were added.

(7) The Discussion section is rather superficial.

Minor points:

(1) Line 61: Many circRNAs are still thought to be noncoding RNAs, e.g. CDR1as.

(2) Line 112: As written, it is unclear if the authors mean the circular RNA is expressed in mouse, if the sequence is conserved in mouse, or both.

(3) Line 114-18: Please be clearer in writing. "RNase R digestion" is not a "circRNA characteristic", resistance to RNase R digestion is a characteristic.

(4) Line 121: "extensive m6A modification". Please use more clear language than "extensive".

(5) Figure 2B: Y-axis title does not make sense as authors are also examining other circRNA expression.

(6) Line 125: Please clarify if MET404 uses the same start codon as the standard MET protein.

(7) Fig 2C: It would be helpful to include in supplemental material a genome browser shot showing the elements (Length? Identity?) that were deleted in the mutant.

Reviewer #4 (Remarks to the Author):

General comment

The manuscript 'Circular RNA-encoded MET variant is a targetable factor in glioblastoma' identifies a circular RNA encoding a 404 amino-acid MET variant (MET404), corresponding to the N-terminal moiety (extracellular Sema domain) of MET. The study investigates the potential role of MET404 as an activator of MET wild-type in glioblastoma stem cells, and as a candidate therapeutic target in glioblastoma experimental models. Identification of MET404 is a remarkable novelty, both for the structure of this naturally-occurring variant, to my knowledge never described before, and the molecular mechanism underlying its expression (circRNA, facilitated by the m6A reader YTHDF2). However, to define the pathogenic impact of MET404, and thus the potential benefits of targeting it with a specific antibody, it is mandatory to robustly investigate the mechanism by which MET404 interacts with and activates wild-type MET. In this respect, the manuscript is currently weak and major points must be addressed.

Major points

1.

The models used to investigate MET404 activities are defined as GSCs (387, 456, 28, 23), kindly provided by a third party (Dr. Jeremy Rich), but no reference or information is provided concerning GSC genetic alterations, transcriptional profile (with subtyping according to the 'classical-mesenchymal-proneural' classification) and expression of tyrosine kinase receptors (to be shown in Western Blot and flow-cytometry) in these cells. It is critical to provide this information to put MET activity in a context. Not only MET amplification is rare, but expression of METwt in glioblastoma is expected to occur preferentially in the mesenchymal subtype and in the absence of EGFR. GSCs without any further specification cannot be considered as universal representative of all glioblastomas. On the contrary, individual GSCs retain the specific properties of the glioblastoma from which they derive, which must be described.

#2.

Although not specified in the methods or result sections, experiments on MET signalling are seemingly performed on GSC cultured in the presence of standard medium (containing EGF and FGF2). Chronic stimulation of EGF and FGF receptors leads to constitutive activation of MAP kinase and PI3-kinase signalling pathways (the two essential pathways downstream MET). All the experiments meant to investigate MET downstream signalling must 1) include analysis of MAP kinase pathway (currently missing); 2) be performed in GSCs kept in a growth-factor free medium for at least 24 hours (the equivalent of serum starvation for conventional cell lines), in order to switch off concomitant signals emanated by EGFR and FGFR.

#3.

If GSC456 and GSC23 express a panel of phosphorylated receptors, including EGFR, PDGFR, FGFR, TRKB, (as shown in Extended Data Figure 4), all well-known to activate AKT signalling, it is surprising that MET404 KD, which abolishes MET phosphorylation only (as shown in Extended Data Figure 4 and in Figure 3b), can downregulate AKT phosphorylation so dramatically (Figure 3b).

#4.

It is unclear whether in GSC28 and GSC387, conveniently used to show the effects of adding exogenous purified MET404, there is no expression of endogenous MET404. If so, it is inferred that MET404 expression is dissociated from expression of METwt. Is MET404 lack of expression in these GSCs due to lack of YTHDF2? As MET is expressed only in a fraction of GBMs (and of corresponding GSCs), and MET404 expression further depends on the inconstant presence of other factors, the overall functional impact of MET404 on GBM might be very limited. Can the authors estimate the frequency of MET404 expression in the overall GBM population?

#5.

In Figure 5i, upper panel, the input lane shows absence of MET expression. How could MET be immunoprecipitated (lane IP-MET)?

#6.

The proposed mechanism by which MET404 should activate METwt is unsound. The authors convincingly map the interaction site between MET404 and METwt in the PSI-IPT domain of METwt. By molecular docking simulation (Fig. 6a), they also show that the site of interaction between MET404 and METwt is unique. A series of important questions arise, that must be addressed experimentally: 1) How can a 'ligand' (MET404) with a single site of interaction with METwt induce the dimerization/oligomerization required to achieve MET activation? Interaction between HGF and MET, leading to MET oligomerization has been attested by several studies, the last of which (Uchikawa et al. Structural basis of the activation of c-MET receptor. *Nat Comm.*, 2021, 12:4074), fully elucidates the mechanism. 2) As MET404 contains the Sema domain, the well-known binding site for HGF, upon MET404 extracellular release only two scenarios seem possible: (i) MET404 competes with METwt for HGF binding, thus acting as a 'decoy', eventually inhibiting MET signalling; (ii) MET404 and METwt form a complex including HGF, which may potentiate MET signalling compared with HGF alone. It is difficult to exclude scenario (i), which would likely occur in vivo, without conceiving scenario (ii). The authors should thoroughly investigate HGF involvement in the METwt-MET404 interaction, using additional model lines expressing the different players and with structural studies. Moreover, the outcomes should be appropriately discussed (current discussion disregards the mechanism of MET activation by MET404).

#7.

The use of onartuzumab in the experimental therapy model is inappropriate. Onartuzumab is known to interfere with MET-HGF binding, but mouse HGF is known to minimally cross-react with human MET. Does GSC456 and GSC23, used for these experiments, express a HGF autocrine loop? Moreover, the efficacy of the antibody combination should be compared with small molecule kinase inhibitors, such as crizotinib, which can effectively block GSC signalling, whatever the upstream MET activation mechanism (and likely more easily crosses BBB compared with antibodies).

Minor points:

#1.

Introduction: as in recent times GBM genetics and its relationship with GBM classification has been refined, by using IDH mutational status as a discriminant between primary GBM and secondary GBM, this should be mentioned where GBM genetics is reported (lines 50-55).

#2.

Introduction/Discussion: the role of YTHDF2 in GSC biology should be more extensively introduced or discussed in light of: Dixit et al., Cancer Discov. 11:480, 2021.

#3.

Page 8, line 160: 'Extended Data Fig. 3h-k' should be corrected in 'Extended Data Fig. 3h-i'. Moreover, the IHC in Extended Data Fig. 3i is poor quality, not convincing.

#4.

Page 51, line 1048: 'The data in c-h' should be corrected in: 'The data in d-h'.

#5.

Extended data Figure 4i: it seems there is an error in graph legend.

Reviewer #5 (Remarks to the Author):

This study identified a circRNA (circMET) in glioblastoma (GBM), and demonstrated its encoding capability, which is driven by m6A modification and YTHDF2 reader. Its coding product (MET404) was shown to initiate GBM by activating MET signalling, which is demonstrated by convincing evidence from genetic mouse model and a series data of in vitro/in vivo experiments. More importantly, this protein was found to bind and activate MET receptor independently of its traditional ligand HGF. Its therapeutic potential was also demonstrated by well-designed in vivo treatments on GBM orthotopic models. Overall, this article is well designed and organized, with excellent clinical relevance and therapeutic potential. Before publication, I have several points for the improvement of the manuscript:

1. Details for bioinformatic data should be improved. The link given seemed to incorporate irrelevant datasets. The accession number for the m6A-seq, RNC-seq and YTHDF2 RIP-seq data should be provided. Furthermore, did the authors investigate circMET in other known databases, such as MiOncoCirc, CSCD2?
2. In Fig1e, the top differentiated genes in RNC-seq should be denoted with gene symbol in the volcano plot for better reading.
3. Line148-149, "Meanwhile, the YTHDF2/circMET interaction was further investigated by IP". To be precise, it should be RNA immunoprecipitation (RIP) instead of IP.
3. The authors reported that circRNA circMET is m6A modified and provided evidence supporting the essential role of its reader YTHDF2 in regulation of circMET translation. However, the upstream mechanism, is lacking. I'd recommend the authors to evaluate the expression of m6A methyltransferase METTL3 in GBM and is there any correlation between METTL3 and m6A modified circMET or MET 404 in GBM?
4. It has been reported that YTHDF2 promotes the degradation of m6A modified mRNAs, here the authors revealed that YTHDF2 promotes the translation of m6A modified circMET? Do the author think the YTHDF2 has distinct roles in regulation of m6A modified mRNA vs circRNA? At least some discussion should be provided.
5. Fig 4e, the IHC staining of indicated antibodies should be quantified using commonly used method.
6. Fig S1a, S2a, the 6 in m6A should be superscript. Please double check the whole manuscript.
7. There are some typos and grammar mistakes that need to be corrected.

MANUSCRIPT ID: NCOMMS-22-22138A-Z

POINT BY POINT RESPONSE TO COMMENTS FROM REVIEWERS

Reviewer #1:

Reviewer 1, Summary: *This manuscript claims the discovery of a new MET protein variant (MET 404) that is encoded by circular MET RNA (circMET) and that is facilitated by the N6 methyladenosine reader YTHDF2. It shows that MET404 promotes glioblastoma (GBM) tumorigenesis by interacting with the MET receptor and promoting MET signalling. Targeting MET404 with a newly developed antibody synergizes with the MET monoclonal antibody onartuzumab in inhibiting in vivo GBM xenograft growth. This appears to be an interesting discovery. However, the experimental plan is severely lacking in rigor and specificity, making the interpretation of the data and the conclusions very questionable. In general, the approaches cannot distinguish between MET404 and MET transcripts and protein as well as previously shown non-coding effects of circMET. Also, the origin of MET404 from circMET is not convincingly established.*

Reviewer 1, Comment 1: *The shRNA used to knock down (KD) circMET is not specific to MET404. It was not tested against linear MET mRNA transcripts. Worse, a quick BLAST alignment by this reviewer revealed that one siRNA aligns with MET mRNA and the other with several other oncogenes. The KD effects that are attributed to MET404 in several data in the manuscript can therefore very well be the resultant of actual MET or other oncogene KD.*

Response:

We sincerely thank the comments. Our current data support the specificity of our circMET shRNAs to MET404. In **Supplementary Fig. 5a** (Extended Data Fig. 4a in the initial version), we have provided the results testing the mentioned

shRNAs on both circular and linear MET transcripts. Our shRNAs specifically target circMET without affecting the expression of its linear counterpart.

Supplementary Fig. 5a. Relative RNA levels of circMET and MET in GSC456/GSC23 cells with or without stable circMET KD.

Of note, the shRNA sequences are all designed to target the unique backsplicing junction (BSJ) of circMET, meaning that they specifically target circMET without binding to MET mRNA. We used two independent RNAi sequences in this study and achieved similar results on knockdown efficiency and downstream phenotypes, further supporting those effects were mediated by the specific targeting to circMET, instead of other off-target silencing.

For an RNAi sequence, the criteria of less than 78% query coverage with other genes and no more than 15 nucleotides matching with the RNA sequence, is well-recognized in siRNA design (*PMID: 26987292*). We also provide the BLAST results here (www.ncbi.nlm.nih.gov/BLAST), showing that none of the circMET RNAi target sequences contains more than 71% query coverage and 15 contiguous base pairs of homologies to other transcripts in human Genomic & transcript database, meeting the requirement of specificity in shRNA design.

[REDACTED]

Blast result for CircMET siRNA1 (5'-GCTTTAATAGGATAAACCTCT-3').

The BLAST results are sorted from high to low. The highest matched gene, ADAM28, only have 71% query coverage (< 78%) and 15 contiguous base pairs (Max Score/2) of homologies to siRNA1 target sequence.

[REDACTED]

Blast result for CircMET siRNA2 (5'- TAATAGGATAAACCTCTCATA-3').

The BLAST results are sorted from high to low. The highest matched gene, MET, only have 71% query coverage (< 78%) and 15 contiguous base pairs (Max Score/2) of homologies to siRNA2 target sequence.

Reviewer 1, Comment 2: Both forward and reverse circMET primers fully align with MET mRNA, creating the same issue described above.

Response:

We sincerely thank the comments. The standard PCR-based method to detect and validate circular RNA is to use a **divergent** primer pair to generate backsplicing junction (BSJ) spanning amplicons (*PMID: 35618955*). Although the divergent primers fully align with linear mRNA, they exclusively amplify circular RNA backsplicing junction because of the **divergent direction (Fig. 2a)**. We provided data showing the circular characteristics of circMET using this divergent primer pair in **Supplementary Fig. 2c-e**, supporting the specificity of these primers to circMET.

Reviewer 1, Comment 3: The antibody that was generated to detect (in several experiments), immunoprecipitate and inhibit MET404 was not shown to be specific for MET404. In fact, supplementary figure 2e and 2g suggest that it actually detects the full MET protein. The interpretation of all data generated using this antibody is therefore questionable.

Response:

We sincerely thank the comments. It is pointed out that the monoclonal antibody for MET404 can also detect the full MET protein; however, when we look at Extended Data Fig. 2e (in the first submitted version, Figure A shown below, uncut blot on its right), apparently the obscure band at the ~100kd is not full-length MET protein (~180kd). And in Extended Data Fig. 2g (in the first submitted version), it's impossible to conclude that MET404 antibody detects the full MET protein (~180kd) from the single band at ~55kd.

A. Left, Coomassie blue-stained purified MET404 with a C-terminal His tag. Right, immunoblot of purified MET404 using the custom anti-MET404 antibody. **B.** Immunoblot for MET404, the membrane marker MET and the cytoplasmic marker β -actin in membrane/cytoplasmic fractionated GSC456 and GSC23 cells. W, whole-cell lysate; C, cytoplasm; M, membrane. **C.** Uncut blot of the repeated experiment of MET404 antibody detecting purified MET404.

MET404 monoclonal antibody was generated against the C-terminus of MET404. The last 4 amino acid “INLS” is unique for MET404, because the translation of circMET spans the backsplicing junction before meeting the stop codon (**Supplementary Fig. 3c**). We also provide additional validation results to confirm the specificity of MET404 antibody using siRNA (**Fig. 2f**) and shRNA (**Fig. 3b**). In the revised manuscript, we repeated the detection of purified MET404 using this monoclonal antibody. A clear band was observed at about 55kd (**Supplementary Fig. 3e**). The uncut blot in the right panel above shows no sign of detecting full-length MET.

Reviewer 1, Comment 4: *CircMET has been discovered and published before and shown to have oncogenic effects that are dependent on its not non-coding effects. Even if the circMET KD effects were specific, the functional consequences could be due to factors other than MET404. The previous publications on circMET are not even cited.*

Response:

We sincerely thank the comments. The function of circMET has been reported to be dependent on its non-coding effects under other pathological conditions (PMID: 32430013, 32614785 and 35042525). Here we

demonstrated that circMET exert its function through its coding ability in GBM and GSCs. The effect of circMET knockdown was rescued by putting back MET404 ORF instead of putting back a mutant circMET RNA (insertion of an A to disrupt the MET404 ORF), demonstrating the effect of circMET in GSCs mainly depends on its coding function (**Supplementary Fig. 5f-p**). Notably, KI mouse model by overexpressing MET404 protein further indicates MET404 is an oncogenic driver in glioblastoma (**Fig. 4**). CircMET biology is heterogenous and of multiple functions. We thank for the reviewer's comment, and we have added these into the revised discussion section.

Reviewer 1, Comment 5: *The demonstration that MET404 is not MET linear splicing variant is not very convincing.*

Response:

We sincerely thank the comments. We have provided two evidence strongly demonstrating the circular origin of MET404. The highest level of evidence for circular RNA translation is (PMID: 35618955): **1)** Detection of endogenous circRNA-derived peptides sensitive to circRNA depletion. **2)** Endogenous protein isoform encoded by circRNA detected by mass spectrometry (MS).

The last 4 amino acid "INLS" of MET404 distinguish itself as a circular translation product from other MET linear splicing variant. The "INLS" is a unique identifier for MET404. Based on this unique identifier, we generated a monoclonal antibody to confirm the existence of MET404, with knockdown validation data available in **Fig. 2f** and **Fig. 3b**. Importantly, we also detected MET404 sequence with the unique "INLS" endogenously in MS (**Supplementary Fig. 3h**). These evidence both strongly support the circular origin of MET404, instead of MET linear splicing variant.

Reviewer #2:

Reviewer 2, Summary: The authors show that a circular RNA from the MET gene is an important player in glioblastoma. It encodes a novel MET variant protein (MET404) that can form a constitutively activated MET receptor whose activity does not require HGF stimulation. The data are intriguing, but I have a number of questions especially with regards to stoichiometry. Some key experimental details are also lacking that makes it difficult to judge the relevance of the conclusions.

Reviewer 2, Comment 1: (1) Figure 1: The authors have used a single annotation algorithm, find_circ, to annotate circular RNAs but it is well established that these algorithms can have high false positive rates. At least one additional annotation algorithm needs to be used, as is common in the field. For non-experts in the circRNA field, it would be helpful to include an additional few sentences on p.4-5 to clarify how circRNAs can be identified using the backsplicing junction sequences.

Response:

We sincerely thank the reviewer for this good question. We applied another algorithm, CIRIquant, to annotate circular RNAs in the RNC-seq and m⁶A-seq dataset. For the m⁶A-seq dataset, circMET (hsa_0082002) is annotated by both algorithms. The Pearson correlation coefficient of the expression level of the intersected circRNAs (recognized by both algorithms) is as high as 0.974 (**Supplementary Fig. 1b**), suggesting highly consistent output of both methods. Similarly, for the RNC-seq dataset, circMET is also included in the intersection. The Pearson correlation coefficient of the circRNAs identified by both methods is as high as 0.993 (**Supplementary Fig. 1b**), also suggesting high consistency. We have also added more details about how circRNAs were identified using the backsplicing junction sequences as the reviewer suggested in the revised manuscripts (in the first paragraph of the results).

Reviewer 2, Comment 2: (2) There is no attention to stoichiometry in this paper. How does the level of circMET compare to the linear MET mRNA? How does the level of the MET404 protein compare to the standard MET protein? The protein data are particularly important given the authors' model for how MET404 functions. The authors should, for example, use an antibody that recognizes both protein isoforms. It is also not clear how the levels of MET404 compare to the MET receptor, and if it is sufficiently high to justify the hyperactivated MET signalling observed in GBM.

Response:

We sincerely thank the reviewer for this insightful comment. We compared the RNA level between circMET and linear MET by Northern blot (**Supplementary Fig. 2j and 5h**). We next investigated the protein levels of MET404. As the reviewer suggested, we ordered an antibody (recognizing the N-terminus of MET) able to detect the full-length MET and MET404 (Abcam, #51067). To validate the band at about 55kd is MET404, we constructed circMET knockdown cell lines. In circMET-KD GSC456 and GSC23 cells, the band at ~55kd was significantly weakened while the other bands remained unchanged (**Supplementary Fig. 4j**). Above data demonstrate this antibody can recognize full length MET and MET404 at ~180kd and ~55kd simultaneously. This antibody was then used to investigate MET and MET404 levels in GSCs. Interestingly, MET404 was found to be expressed at comparable or even higher level as compared to MET in some of the GSCs and GBM tissues (**Fig. 2j and Supplementary Fig. 4k**). The high expression of MET404 may be attributed to the longer half-life of circMET (**Supplementary Fig. 2e**). In addition, the multiple m⁶A modification on circMET and the preferential expression of YTHDF2 in GSC and GBM may also contribute to the high protein level of MET404 by driving the efficient translation of circMET. The above data suggest MET404, encoded by circMET, is highly expressed and a more promising therapeutic target than MET in GBM.

Reviewer 2, Comment 3: (3) Northern blots are a standard approach for proving/validating high circular RNA expression, but these are lacking in the manuscript.

Response:

We sincerely thank the reviewer for this good question. We synthesized exon- and junction-specific probes and performed Northern blotting using a NorthernMax-Gly kit (Invitrogen Cat. AM1946) and a biotin-labelled probe detection kit (Beyotime Cat. GS009), validating high expression of circMET in indicated GSC cell lines (**Supplementary Fig. 2j**). This method was also used to validate successful overexpression of circMET in MES28 (**Supplementary Fig. 5h**, also for Comment 5 below).

Reviewer 2, Comment 4: (4) Figure 2I, J: Is the standard MET protein also over-expressed in these particular cancerous tissues?

Response:

We sincerely thank the reviewer for this good question. We re-evaluated the expression level of MET in these paired tissues (45 pairs still in stock) and semi-quantified them by immunoblot as was done previously in the manuscript. MET overexpression is seen in ~48.9% of the samples (**Fig. 2i and Supplementary 4i**, or see below, Figure A and B). We also re-evaluated p-MET and MET404 levels in these paired tissues. Co-upregulation of MET404 and p-MET was observed in 75.6% of the patients, while co-upregulation of MET and p-MET was seen in 35.6% of the patients (see below, Figure C and D). In cancerous tissues alone, MET404 better correlates with p-MET than MET (**Fig. 7b**), demonstrating the essential role of MET404 in MET signalling activation. These data together suggest that MET404 upregulation is commonly seen and is a better predictor of MET signalling activity than MET receptor in GBM.

A. Protein levels of MET404, MET, pMET in randomly selected peritumor and tumor tissues of GBM patients from the cohort in Fig. 2i. **B-D.** Scatter plot showing the ratio of indicated protein expression levels between tumor (T) and peritumor (P) tissue. n=45.

Reviewer 2, Comment 5: (5) Figure 3A: The authors have over-expressed circMET, but no validation is provided to show that the circRNA was over-expressed or how cleanly it was done, e.g. are linear RNAs also over-expressed. This is best done using Northern blots. There are also no details explaining how the over-expression approach works. Sufficient details need to be provided in the methods so that the work can be replicated by a reader if they desired to do so.

Response:

We sincerely thank the reviewer for this good question. In the initially submitted manuscript, we validated the overexpression of circMET by qPCR and immunoblot (revised **Supplementary Fig. 5g and 5i**; 4g and 4h in the initial manuscript). These data indicated that circMET was successfully overexpressed while the level of linear MET remained unchanged. We also used Northern blot to validate the overexpression as suggested (**Supplementary Fig. 5h**). We applied lentivirus to overexpress the circRNA.

Constructs of circMET, circMET Mut, circMET ORF was co-transfected with virus packaging plasmids in 293T cells to produce overexpress lentivirus. Details of the overexpression approach has been added to the revised manuscript (see methods, line 483).

Reviewer 2, Comment 6: (6) Fig 6: Addition of MET404 protein is used, but it is unclear if physiologically relevant levels of purified protein were added.

Response:

We sincerely thank the reviewer for this excellent question. In the initially submitted manuscript, we performed *in vitro* gradient stimulation and confirmed 1µg/mL of the self-purified MET404 best activated MET signalling, and we achieved similar effects using this concentration as compared to 100 ng/mL of commercial purified HGF, which is a commonly used concentration for HGF treatment (*PMID: 32736659, 28423312, 11408346*) (**Supplementary Fig. 6d**). MET signalling can also be activated with a low concentration (100 ng/mL) of MET404.

In addition, we ordered another clone of MET404 monoclonal antibody and applied it as capture antibody to measure *in vivo* MET404 level in cerebrospinal fluid (CSF) from a cohort of GBM patients using a sandwich ELISA protocol. MET404 level fluctuates at 711.2 ± 217.5 pg/mL in GBM CSF (**Supplementary Fig. 6c**), similar to reported HGF CSF level (range: 207-893 pg/mL, *PMID: 11396995, 19856144*).

Reviewer 2, Comment 7: (7) The Discussion section is rather superficial.

Response:

We sincerely thank the reviewer for this comment. We summarized the suggestions from all the reviewers and refined our discussion in the manuscript. We discussed more on the distinct function of circMET and MET404 in GBM

biology, the critical role of YTHDF2 in facilitating circRNA translation, the mechanism of MET404 activating MET and the reason why better efficacy was observed in antibody combination rather than TKIs.

Reviewer 2, Comment 8: Minor points: (1) Line 61: Many circRNAs are still thought to be noncoding RNAs, e.g. CDR1as.

Response:

We sincerely apologize for this imprecise statement. In the revised manuscript, we have added some words to avoid misunderstanding.

Reviewer 2, Comment 9: (2) Line 112: As written, it is unclear if the authors mean the circular RNA is expressed in mouse, if the sequence is conserved in mouse, or both.

Response:

We sincerely apologize for this imprecise writing. CircMET is highly conserved across some species (human, mouse, rat and macaca). It is expressed in mice, formed from exon3 of murine *MET*. We have refined the sentences in the revised manuscript.

Reviewer 2, Comment 10: (3) Line 114-18: Please be clearer in writing. "RNase R digestion" is not a "circRNA characteristic", resistance to RNase R digestion is a characteristic.

Response:

We sincerely apologize for this mistake. The sentences have been rewritten in the revised manuscript.

Reviewer 2, Comment 11: (4) Line 121: “extensive m6A modification”. Please use more clear language than “extensive”.

Response:

We sincerely apologize for this imprecise writing. In the revised manuscript, we have changed the word to “multiple”.

Reviewer 2, Comment 12: (5) Figure 2B: Y-axis title does not make sense as authors are also examining other circRNA expression.

Response:

We sincerely apologize for this mistake. We have changed the title to “relative RNA enrichment” in the revised figures.

Reviewer 2, Comment 13: (6) Line 125: Please clarify if MET404 uses the same start codon as the standard MET protein.

Response:

We sincerely thank the reviewer for this suggestion. MET404 indeed uses the same start codon as the standard MET protein, but its ORF generates a unique C-terminus. Details have been added in the revised manuscript.

Reviewer 2, Comment 14: (7) Fig 2C: It would be helpful to include in supplemental material a genome browser shot showing the elements (Length? Identity?) that were deleted in the mutant.

Response:

We sincerely thank the reviewer for this suggestion. We have added the illustration as suggested (**Supplementary Fig. 4a**).

Reviewer #4:

Reviewer 4, Summary: The manuscript 'Circular RNA-encoded MET variant is a targetable factor in glioblastoma' identifies a circular RNA encoding a 404 amino-acid MET variant (MET404), corresponding to the N-terminal moiety (extracellular Sema domain) of MET. The study investigates the potential role of MET404 as an activator of MET wild-type in glioblastoma stem cells, and as a candidate therapeutic target in glioblastoma experimental models. Identification of MET404 is a remarkable novelty, both for the structure of this naturally-occurring variant, to my knowledge never described before, and the molecular mechanism underlying its expression (circRNA, facilitated by the m6A reader YTHDF2).

However, to define the pathogenic impact of MET404, and thus the potential benefits of targeting it with a specific antibody, it is mandatory to robustly investigate the mechanism by which MET404 interacts with and activates wild-type MET. In this respect, the manuscript is currently weak and major points must be addressed.

Reviewer 4, Comment 1: Major points # 1. The models used to investigate MET404 activities are defined as GSCs (387, 456, 28, 23), kindly provided by a third party (Dr. Jeremy Rich), but no reference or information is provided concerning GSC genetic alterations, transcriptional profile (with subtyping according to the 'classical-mesenchymal-proneural' classification) and expression of tyrosine kinase receptors (to be shown in Western Blot and flow-cytometry) in these cells. It is critical to provide this information to put MET activity in a context. Not only MET amplification is rare, but expression of METwt in glioblastoma is expected to occur preferentially in the mesenchymal subtype and in the absence of EGFR. GSCs without any further specification cannot be considered as universal representative of all glioblastomas. On the contrary, individual GSCs retain the specific properties of the glioblastoma from which they derive, which must be described.

Response:

We sincerely thank the reviewer for this excellent comment. Here we provide the requested information for the GSCs used in the manuscript (also included in the **Supplementary Data**):

1) Genetic alteration: We incorporated the data from two published reports (*JEM*, PMID: 30948495 and *Cell Metabolism*, PMID: 33406399) and summarized the genetic information of a panel of 47 GSCs, including the GSCs used in the manuscript. *EGFR* and *PDGFRA* amplifications are both seen in GSC456 while GSC387 only harbours *EGFR* amplification. *MET* amplification or deletion are both absent in the GSCs tested. The other genetic alterations are illustrated below:

2) Transcriptional profile: We used a well-recognized classifier to infer the molecular subtypes (Proneural, Classical and Mesenchymal) of these GSCs (PMID: 28697342) based on their transcriptome. Briefly, single sample gene set enrichment analysis ssGSEA-based equivalent distribution resampling classification strategy was performed based on 50-gene signatures in each subtype. Three empirical classification P values for three subtypes were generated for each sample. The molecular subtype for each sample was defined by the smallest P value. As the table below shows, GSCs used in the manuscript covered all the subtypes of glioblastoma, including classical

(GSC387, GSC23), proneural (GSC28), mesenchymal (GSC456), avoiding subtype bias.

Glioblastoma Stem Cell	Patient Age (Years), Gender	Transcriptional Subtype	Tumor Grade
GSC387	76 years, Female	Classical	Glioblastoma (Grade IV)
GSC456	8 years, Female	Mesenchymal	Glioblastoma (Grade IV)
GSC28	Unknown	Proneural	Glioblastoma (Grade IV)
GSC23	63 years, Male	Classical	Recurrent Glioblastoma (Grade IV)

3) Tyrosine kinase receptors expression: We used Western blot to detect RTK and their phosphorylation levels in these GSCs. Consistent with their transcriptional subtype, GSC456 expresses a higher level of MET, while high expression of EGFR is seen in GSC23 and GSC387. The expression levels are shown below.

Reviewer 4, Comment 2: Major points # 2. Although not specified in the methods or result sections, experiments on MET signalling are seemingly performed on GSC cultured in the presence of standard medium (containing EGF and FGF2). Chronic stimulation of EGF and FGF receptors leads to constitutive activation of MAP kinase and PI3-kinase signalling pathways (the two essential pathways downstream MET). All the experiments meant to investigate MET downstream signalling must 1) include analysis of MAP kinase pathway (currently missing); 2) be performed in GSCs kept in a growth-factor free medium for at least 24 hours (the equivalent of serum starvation for conventional cell lines), in order to switch off concomitant signals emanated by EGFR and FGFR.

Response:

We sincerely thank the reviewer for this excellent suggestion.

1) We repeated the experiments in the concerning panel and included detection of the phospho-ERK level to evaluate MAPK signalling and found this pathway changed as the way PI3K-Akt does (**Fig. 3, Fig. 5-7, Supplementary Fig. 5-7**).

2) All the experiments meant to investigate MET downstream signalling in the revised manuscript were done using GSCs kept in medium free of EGF and bFGF for 24 hours. The details have also been specified in the methods (line 463).

Reviewer 4, Comment 3: Major points #3. If GSC456 and GSC23 express a panel of phosphorylated receptors, including EGFR, PDGFR, FGFR, TRKB, (as shown in Extended Data Figure 4), all well-known to activate AKT signalling, it is surprising that MET404 KD, which abolishes MET phosphorylation only (as shown in Extended Data Figure 4 and in Figure 3b), can downregulate AKT phosphorylation so dramatically (Figure 3b).

Response:

We sincerely thank the reviewer for this good question. MET404 KD in GSC456 and GSC23 dramatically reduced the phosphorylation of AKT, indicating the MET signalling is one of the dominant RTK pathways for GSC456 and GSC23 maintenance. As shown in **Supplementary Data**, GSC456 and GSC23 showed marked activation of MET phosphorylation among the GSCs, suggesting the essential role of MET signalling in GSC456 and GSC23 maintenance. In contrast, GSC387 exhibited a low level of p-MET and a high level of p-EGFR, indicating EGFR signalling is likely to serve as the dominant RTK pathway in GSC387. To further verify this, we employed GSC387 with MET or EGFR inhibition as a control. With similar knockdown levels, loss of EGFR dramatically reduced AKT phosphorylation, while MET404 knockdown only

mildly inhibited downstream p-AKT (see below). Collectively, our results demonstrate the crucial role of MET404 and MET signalling activity in those GSCs with high baseline p-MET levels.

Reviewer 4, Comment 4: Major points #4. It is unclear whether in GSC28 and GSC387, conveniently used to show the effects of adding exogenous purified MET404, there is no expression of endogenous MET404. If so, it is inferred that MET404 expression is dissociated from expression of METwt. Is MET404 lack of expression in these GSCs due to lack of YTHDF2? As MET is expressed only in a fraction of GBMs (and of corresponding GSCs), and MET404 expression further depends on the inconstant presence of other factors, the overall functional impact of MET404 on GBM might be very limited. Can the authors estimate the frequency of MET404 expression in the overall GBM population?

Response:

We sincerely thank the reviewer for this good question. The absence of MET404 in GSC28 and GSC387 was due to the short exposure time in the previous immunoblot. We repeated the immunoblot to detect the MET404 expression in those GSCs (**Supplementary Fig. 4I**). GSC28 and GSC387 do express MET404, but at a lower level compared to GSC456 and GSC23, consistent with the circMET RNA levels (**Supplementary Fig. 2g**).

A lower level of YTHDF2 was also seen in GSC28/GSC387 as compared to GSC456/23 (see below). CircMET RNA and YTHDF2 level together result in

lower expression of MET404 in GSC28/387. YTHDF2 and MET404 expression are also spatially correlated in GBM slides as validated by IHC (Fig. 2k). Overall, these findings suggest that MET404 production depends not only on circMET (generated from MET pre-mRNA) at transcriptional level but also on YTHDF2 at translational level.

MET404 is pervasively expressed in GBM rather than restricted to a limited subset, as observed in nearly all the randomly chosen GBM samples in the IHC staining (IHC score>0: ~89.2%) (Fig. 2i, k, and see below). In addition, MET404 has a strong correlation with phospho-MET level, despite absence of MET upregulation (Fig. 7b, Fig. 2i and Supplementary Fig. 4i, see also response for comment 2 from Reviewer #2). Overall, these data indicate the pronounced role of MET404 in activating MET signalling and thus promoting GBM progression.

A. Protein levels of MET404 and YTHDF2 in the indicated GSCs. **B.** IHC score of MET404 expression. Data are presented as box plot containing the median (center line), the first and third quartiles (box limits). The whiskers indicate the maxima and the minima. **C-E.** Representative IHC images showing different levels (low to high, from left to right) of MET404 expressions in slides of GBM patients from Fig. 2k.

Reviewer 4, Comment 5: Major points #5. In Figure 5i, upper panel, the input lane shows absence of MET expression. How could MET be immunoprecipitated (lane IP-MET)?

Response:

We sincerely thank the reviewer for this good question. We checked the experiment records and believed that the blurred input lane was due to insufficient loading of input protein. Below (Figure A) is the long exposure mode of the same blot in the initially submitted Fig. 5i. MET expression can be identified in the input lane.

We repeated the experiment again with refined conditions. The updated data has been put into the revised **Fig. 5** (Figure B, above).

Reviewer 4, Comment 6: Major points #6. The proposed mechanism by which MET404 should activate METwt is unsound. The authors convincingly map the interaction site between MET404 and METwt in the PSI-IPT domain of METwt. By molecular docking simulation (Fig. 6a), they also show that the site of interaction between MET404 and METwt is unique. A series of important questions arise, that must be addressed experimentally: 1) How can a 'ligand' (MET404) with a single site of interaction with METwt induce the dimerization/oligomerization required to achieve MET activation? Interaction between HGF and MET, leading to MET oligomerization has been attested by several studies, the last of which (Uchikawa et al. Structural basis of the activation of c-MET receptor. Nat Comm., 2021, 12:4074), fully elucidates the mechanism. 2) As MET404 contains the Sema domain, the well-known binding

site for HGF, upon MET404 extracellular release only two scenarios seem possible: (i) MET404 competes with METwt for HGF binding, thus acting as a 'decoy', eventually inhibiting MET signalling; (ii) MET404 and METwt form a complex including HGF, which may potentiate MET signalling compared with HGF alone. It is difficult to exclude scenario (i), which would likely occur in vivo, without conceiving scenario (ii). The authors should thoroughly investigate HGF involvement in the METwt-MET404 interaction, using additional model lines expressing the different players and with structural studies. Moreover, the outcomes should be appropriately discussed (current discussion disregards the mechanism of MET activation by MET404).

Response:

We sincerely thank the reviewer for these excellent questions.

1) First, the interaction sites on MET404 predicted by molecular docking span from 42 to 358 (**Fig. 6a**), which indicates the broad interaction interface between MET404 and MET receptor rather than a single site interaction. Although we confirmed the importance of amino acid Q328/D358 in the interaction by mutagenesis analysis (**Supplementary Fig. 7a**), these results did not exclude the participation of other sites. Indeed, nearly all the sequence of MET404 might interact with MET receptor, providing the theoretical evidence for MET receptor dimerization.

Second, we here provide two strongly supportive experimental evidence to prove that MET404 can arise the dimerization of MET receptor:

a) To exclude dimerization arisen by endogenously secreted HGF, we established 293T HGF-KO cells by CRISPR-Cas9 technology (**Supplementary Fig. 7e**). We synthesized two constructs to overexpress MET monomer with a flag or a HA tag. If MET dimers form after addition of MET404, an HA antibody can immunoprecipitate flag-tagged MET, or vice versa. As the data in **Fig. 6e** shows, in control cells (293T HGF KO), HA-tagged and flag-tagged MET interacted with each other, probably mediated by endogenous MET404. After

addition of MET404, the interaction intensified between these differently tagged MET monomers.

b) We utilized a commercial kit (NanoBit Protein-Protein Interaction system, Promega #N2014), which is also used in a published report to investigate dimerization (*PMID: 31474362*), to monitor dimerization after addition of purified MET404 to the cells. When dimerization occurs, the approximated residues (LgBit and SmBit) fused on the C-terminus of both monomers generate a luminescent signal. After adding MET404, we recorded a sudden rise of luminescent signal in 10 minutes, suggesting the rapid formation of MET dimers under MET404 stimulation. The signal sustained for over half an hour, while the control cells remained underactive (**Fig. 6f**).

In summary, we provided two evidence to prove the ability of MET404 in inducing dimerization of MET receptor in HGF KO cells, further supporting MET404 alone can activate MET signalling. The detailed structural basis underlying the process may be elucidated in future studies.

2) First, in theory, MET404 is unlikely to interact with HGF. The SEMA domain, a seven-blade-like structure, is formed by N-terminal amino acids (25-514) of linear MET. *Uchikawa et al.* reported that the interaction between HGF and MET SEMA domain relies on four key interfaces. These interfaces are made between different blades of SEMA and domains of HGF, as shown below:

	Blades on SEMA	Domains on HGF
Interface I	Blades 5, 6	two short inter-strand loops in the N domain
Interface II	Blades 4E, 4F	K2 domain
Interface III	Blades 4E, 5A, 5B, 6C, 6D	K3 domain
Interface IV	Blades 2, 3	SPH domain

The capitals denote different strands on the blades (e.g., blade 4E means strand E of blade 4).

Summarized from *E. Uchikawa et al., Nat Commun. 12, 4074 (2021). PMID: 34210960*

The graph below is adapted from a well-known report (*PMID: 15167892*) on the crystal structure of MET SEMA domain. As illustrated below, **the sequence of MET404 only spans blade 1, 2, 3, 4A-4D, 5, 7D.**

[REDACTED]

The reported key interfaces (*PMID: 34210960*) between SEMA domain and HGF mainly require blade 5A, 5B, 4E, 4F, 6, **the latter three of which are absent from MET404.** Blades 2-3 only form a low-affinity binding, according to the authors (*PMID: 15167892*). Since MET404 does not possess complete sequence of SEMA domain and lacks key region mediating interaction with HGF reported by these structural studies, it seems unable, in theory, to interact with HGF in physiological settings.

Second, we provide experimental evidence to support the inability of MET404 to interact with HGF. We performed Co-IP analysis in 293T MET-KO cell lines (**Supplementary Fig. 7h**) transfected with tagged HGF and MET404 constructs, where loss of MET rule out the possibility of any indirect binding between HGF and MET404 bridged by MET. We observed no interaction

between HGF and MET404 in the immunoprecipitation experiment (long exposure mode used, **Supplementary Fig. 7i**). These data suggest that MET404 and HGF do not interact with each other but instead they are able to independently bind to different sites of MET and result in dimerization and activation of MET receptor. We have also refined the discussion part under the reviewer's suggestion.

Reviewer 4, Comment 7: Major points #7. The use of onartuzumab in the experimental therapy model is inappropriate. Onartuzumab is known to interfere with MET-HGF binding, but mouse HGF is known to minimally cross-react with human MET. Does GSC456 and GSC23, used for these experiments, express a HGF autocrine loop? Moreover, the efficacy of the antibody combination should be compared with small molecule kinase inhibitors, such as crizotinib, which can effectively block GSC signalling, whatever the upstream MET activation mechanism (and likely more easily crosses BBB compared with antibodies).

Response:

We sincerely thank the reviewer for these excellent questions. GSCs have been reported to express an HGF autocrine loop (PMID: 32642717). The supernatant of GSC456 and GSC23 was collected and subjected to immunoblot analysis. HGF is detected in the supernatant (see below, Figure A). By using mass spectrometry, we also verified the existence of HGF in the supernatant (see below, Figure B, C). Knockout of HGF in these cells impairs MET signalling (**Supplementary Fig. 7d**). IF staining of GSC456/23 xenografts shows the colocalization of human HGF and MET *in vivo*, consistent with the *in vitro* validation (see below, Figure D). These results support that GSC456 and GSC23 are able to secrete HGF and express MET receptor themselves, thus creating a local HGF autocrine loop.

We also compared the efficacy of combined antibody therapy and TKI. GSC456 and GSC23 was intracranially implanted. The antibodies were given orthotopically (*i.c.*) into the tumour site 10 days after implantation as was done previously (**Fig. 7e**), while crizotinib were given from this time point via oral gavage (*i.g.*, 50mg/kg) every other day as previously reported (*PMID*: 31694905). Control mice were given PBS orthotopically (*i.c.*) or the solvent (*i.g.*). It appears that antibody combination constrained the growth of tumors and extended the survival more effectively than crizotinib in both GSC-bearing mice (**Supplementary Fig. 8a, b**). RTKs may develop mutations that reduce binding capacity of TKI during treatment (*PMID*: 36115852), resulting in compromised therapeutic efficacy, while antibody combination therapy (targeting ligands but not the receptor) can circumvent this intractability. This may be why the efficacy of antibody combination exceeded crizotinib. Collectively, these results suggest the promising efficacy of the antibody combination strategy in repressing the activation MET signalling and GBM malignancy.

A. Immunoblot of concentrated supernatant from the culture medium of GSC456 and GSC23 with the HGF siRNAs or scramble RNA treatment. Coomassie blue-stained total proteins were used as a loading control. **B.** Mascot search results of MS analysis revealed that HGF sequences identified in the concentrated supernatant of GSC456. **C.** detection of HGF amino acid sequences by MS. **D.** IF images showing MET and HGF expression and colocalization from GSC456 or GSC23 xenografts slides. Scale bar, 30µm.

Reviewer 4, Comment 8: Minor points #1. Introduction: as in recent times GBM genetics and its relationship with GBM classification has been refined, by using IDH mutational status as a discriminant between primary GBM and secondary GBM, this should be mentioned where GBM genetics is reported (lines 50-55).

Response:

We sincerely appreciate this valuable comment. The relevant content has been added in the revised manuscript.

Reviewer 4, Comment 9: Minor points #2. Introduction/Discussion: the role of YTHDF2 in GSC biology should be more extensively introduced or discussed in light of: Dixit et al., Cancer Discov. 11:480, 2021.

Response:

We sincerely appreciate this valuable comment. YTHDF2 is reported to promote the degradation of m⁶A-modified mRNAs; however, in GSCs of GBM, it protects mRNA of MYC and VEGFA from degradation in an m⁶A-dependent manner. Our data also supports this distinct role in GBM, for it also participated in enhancing the translation of tumour-promoting circRNA (circMET) with m⁶A modifications. It will be interesting to investigate mechanism driving this tissue-specific function in future studies. More discussion has been added in the revised manuscript.

Reviewer 4, Comment 10: Minor points #3. Page 8, line 160: 'Extended Data Fig. 3h-k' should be corrected in 'Extended Data Fig. 3h-i'. Moreover, the IHC in Extended Data Fig. 3i is poor quality, not convincing.

Response:

We sincerely thank the reviewer. The Figure numbers should be correct in the revised manuscript. It appears that Extended Data Fig. 3i (revised

Supplementary Fig. 4n) is not an IHC image. We rechecked the data quality of IHC images, and repeated the experiment in Extended Data Fig. 4m and 4o and gained more convincing results as shown in revised **Supplementary Fig. 5n and 5p**.

Reviewer 4, Comment 11: Minor points #4. Page 51, line 1048: 'The data in c-h' should be corrected in: 'The data in d-h'.

Response:

We sincerely apologize for this mistake. It has been corrected in the revised version.

Reviewer 4, Comment 12: Minor points #5. Extended data Figure 4i: it seems there is an error in graph legend.

Response:

We sincerely apologize for this mistake. It has been refined in the revised version.

Reviewer #5:

Reviewer 5, Summary: *This study identified a circRNA (circMET) in glioblastoma (GBM), and demonstrated its encoding capability, which is driven by m6A modification and YTHDF2 reader. Its coding product (MET404) was shown to initiate GBM by activating MET signalling, which is demonstrated by convincing evidence from genetic mouse model and a series data of in vitro/in vivo experiments. More importantly, this protein was found to bind and activate MET receptor independently of its traditional ligand HGF. Its therapeutic potential was also demonstrated by well-designed in vivo treatments on GBM orthotopic models. Overall, this article is well designed and organized, with excellent clinical relevance and therapeutic potential. Before publication, I have several points for the improvement of the manuscript:*

Reviewer 5, Comment 1: *Details for bioinformatic data should be improved. The link given seemed to incorporate irrelevant datasets. The accession number for the m6A-seq, RNC-seq and YTHDF2 RIP-seq data should be provided. Furthermore, did the authors investigate circMET in other known databases, such as MiOncoCirc, CSCD2?*

Response:

We sincerely thank the reviewer for this suggestion. The accession IDs for the sequencing dataset have been updated in the revised manuscript. In addition to circBase, we also compare CircMET with other cancer circRNA database as the reviewer suggested, showing consistency of CircMET among different databases:

(1) MiOncoCirc: The genomic position of CircMET in this database (chr7:116699070|116700284) is different from that in circBase (chr7:116339124-116340338) because of the use of a different version of the genome. The sequence is identical. However, there appear not to be sufficient GBM samples included in this database.

[REDACTED]

(2) CSCD2: The genomic position of CircMET in this database is the same as that in MiOncoCirc. The sequence, the genomic and spliced lengths are identical to those in circBase, showing consistency of CircMET between these databases.

[REDACTED]

Reviewer 5, Comment 2. *In Fig1e, the top differentiated genes in RNC-seq should be denoted with gene symbol in the volcano plot for better reading.*

Response:

We sincerely thank the reviewer for this suggestion. We added some labels without spoiling readability of the graph.

Reviewer 5, Comment 3. Line148-149. “Meanwhile, the YTHDF2/circMET interaction was further investigated by IP”. To be precise, it should be RNA immunoprecipitation (RIP) instead of IP.

Response:

We sincerely apologize for this mistake. It has been corrected in the revised manuscript.

Reviewer 5, Comment 4. The authors reported that circRNA circMET is m6A modified and provided evidence supporting the essential role of its reader YTHDF2 in regulation of circMET translation. However, the upstream mechanism, is lacking. I'd recommend the authors to evaluate the expression of m6A methyltransferase METTL3 in GBM and is there any correlation between METTL3 and m6A modified circMET or MET 404 in GBM?

Response:

We sincerely thank the reviewer for this excellent suggestion. We knocked down METTL3 in GSCs specifically by siRNAs and observed unchanged levels of circMET by qPCR (see below, left), but the protein level of MET404 was downregulated (**Supplementary Fig. 4f**, or see below, right), consistent with the result of overexpressing m⁶A eraser ALKBH5 (**Supplementary Fig. 4g**). This further demonstrates circMET translation is m⁶A-driven. However, the role of METTL3 in GBM (or GSCs) is still not well-characterized. It has been reported to suppress the growth and self-renewal of GSCs (*PMID: 28297667*), while it promotes the stemness of GSCs and its resistance to TMZ in another report (*PMID: 34336690*). Our data suggested that the well-illustrated GBM marker, YTHDF2, works as the main regulator of circMET translation. The involvement of m⁶A writers METTL3/METTL14 and the detailed underlying mechanism in circRNA translation may need further investigation.

Reviewer 5, Comment 5. It has been reported that YTHDF2 promotes the degradation of m6A modified mRNAs, here the authors revealed that YTHDF2 promotes the translation of m6A modified circMET? Do the author think the YTHDF2 has distinct roles in regulation of m6A modified mRNA vs circRNA? At least some discussion should be provided.

Response:

We sincerely thank the reviewer for this excellent question. Although YTHDF2 has been reported to enhance the degradation of mRNA with m⁶A modifications, it may have a unique role in GBM. Not only our data implies this, the excellent work by Dixit et al. also suggested its tissue-specific role in maintaining stability of VEGF and MYC mRNA, as opposed to previously proved function in promoting mRNA degradation. From this perspective, these markers (YTHDF2 or MET404 in this manuscript), with marked role in GBM, may have great potential to serve as targets for GBM precise treatment. We have discussed more on this in the revised manuscript under the reviewer's kind suggestion.

Reviewer 5, Comment 6: Fig 4e, the IHC staining of indicated antibodies should be quantified using commonly used method.

Response:

We sincerely thank the reviewer for this suggestion. The quantified data has been added in the revised Fig. 4.

Reviewer 5, Comment 7: Fig S1a, S2a, the 6 in m6A should be superscript.
Please double check the whole manuscript.

Response:

We sincerely apologize for this mistake. We have checked through the manuscript again to ensure such mistakes have been corrected.

Reviewer 5, Comment 8: There are some typos and grammar mistakes that need to be corrected.

Response:

We sincerely apologize for these mistakes. The revised manuscript has been checked under the help of two native English speakers to minimize the appearance of those mistakes.

REVIEWERS' COMMENTS

Reviewer #1 (Remarks to the Author):

The authors satisfactorily addressed my critique and alleviated my previous concerns about the shRNA, primers and antibody.

Reviewer #3 (Remarks to the Author):

The authors have addressed most of my prior concerns. I only have several suggestions for minor additions to the manuscript:

- The authors need to provide full details for how the circRNA overexpression plasmids were generated. For example, they have inserted an A into the ORF to disrupt translation but the manuscript does not provide the exact location of the insertion. It is also not clear how they changed the pCDH-CMV-MCS-EF1-copGFP-T2A-Puro vector so that it produces a circRNA, e.g. what intronic sequences were used to drive backsplicing? Please provide full sequences of all gene fragments that were synthesized. Such details are critical for ensuring clarity and reproducibility of results.
- Supp Fig 3g: Please show the entire blot to prove the specificity of the MET404 antibody in cell lysates.
- Line 40: Some circRNAs are noncoding RNAs so it is inappropriate to say they were "previously recognized as noncoding RNAs".

Reviewer #4 (Remarks to the Author):

The authors conducted appropriate experiments to diligently address my criticisms, thereby significantly improving the manuscript.

However, I believe that the manuscript convincingly demonstrates that only GBMs exhibiting constitutive activation of the MET tyrosine kinase receptor can respond to a combination therapy that includes a 'traditional' MET inhibitor and a neutralizing antibody targeting MET404. Given the frequency of MET expression in GBM (relevant only in the mesenchymal subtype, approx. 35% of all GBMs) and the even lower percentage of cases with MET constitutive activation, this limitation of the study, which is noted in the Discussion, should also be clearly stated in the Abstract. Proper patient identification for clinical trials is crucial for evaluating the therapeutic response.

Reviewer #5 (Remarks to the Author):

The authors have well addressed my comments. I have no other comment.

POINT BY POINT RESPONSE TO COMMENTS FROM REVIEWERS
(Final Revision)

Reviewer #1 (Remarks to the Author):

The authors satisfactorily addressed my critique and alleviated my previous concerns about the shRNA, primers and antibody.

Reviewer #3 (Remarks to the Author):

The authors have addressed most of my prior concerns. I only have several suggestions for minor additions to the manuscript:

- The authors need to provide full details for how the circRNA overexpression plasmids were generated. For example, they have inserted an A into the ORF to disrupt translation but the manuscript does not provide the exact location of the insertion. It is also not clear how they changed the pCDH-CMV-MCS-EF1-copGFP-T2A-Puro vector so that it produces a circRNA, e.g. what intronic sequences were used to drive backsplicing? Please provide full sequences of all gene fragments that were synthesized. Such details are critical for ensuring clarity and reproducibility of results.

Response:

We sincerely appreciate the reviewer for this suggestion. Full sequences of the gene fragments used for circRNA overexpression have been provided in the Supplementary Information.

- Supp Fig 3g: Please show the entire blot to prove the specificity of the MET404 antibody in cell lysates.

Response:

We sincerely appreciate the reviewer for this suggestion. The blot has been

added to **Supplementary Fig. 3e**. The uncut gels and blots are also provided in the Source Data file.

- Line 40: Some circRNAs are noncoding RNAs so it is inappropriate to say they were "previously recognized as noncoding RNAs".

Response:

We sincerely appreciate the reviewer for this suggestion. The statement has been refined in the revised manuscript.

Reviewer #4 (Remarks to the Author):

The authors conducted appropriate experiments to diligently address my criticisms, thereby significantly improving the manuscript.

However, I believe that the manuscript convincingly demonstrates that only GBMs exhibiting constitutive activation of the MET tyrosine kinase receptor can respond to a combination therapy that includes a 'traditional' MET inhibitor and a neutralizing antibody targeting MET404. Given the frequency of MET expression in GBM (relevant only in the mesenchymal subtype, approx. 35% of all GBMs) and the even lower percentage of cases with MET constitutive activation, this limitation of the study, which is noted in the Discussion, should also be clearly stated in the Abstract. Proper patient identification for clinical trials is crucial for evaluating the therapeutic response.

Response:

We sincerely appreciate the reviewer for this suggestion. The abstract has been revised as required.

Reviewer #5 (Remarks to the Author):

The authors have well addressed my comments. I have no other comment.